# Interaction-Aware Gaussian Weighting for Clustered Federated Learning

**Alessandro Licciardi** [* 1 2]   **Davide Leo** [* 3]   **Eros Fanì** [3 4]   **Barbara Caputo** [3]   **Marco Ciccone** [† 5]

## Abstract

Federated Learning (FL) emerged as a decentralized paradigm to train models while preserving privacy. However, conventional FL struggles with data heterogeneity and class imbalance, which degrade model performance. Clustered FL balances personalization and decentralized training by grouping clients with analogous data distributions, enabling improved accuracy while adhering to privacy constraints. This approach effectively mitigates the adverse impact of heterogeneity in FL. In this work, we propose a novel clustered FL method, `FedGWC` (Federated Gaussian Weighting Clustering), which groups clients based on their data distribution, allowing training of a more robust and personalized model on the identified clusters. `FedGWC` identifies homogeneous clusters by transforming individual empirical losses to model client interactions with a Gaussian reward mechanism. Additionally, we introduce the *Wasserstein Adjusted Score*, a new clustering metric for FL to evaluate cluster cohesion with respect to the individual class distribution. Our experiments on benchmark datasets show that `FedGWC` outperforms existing FL algorithms in cluster quality and classification accuracy, validating the efficacy of our approach. Code is available at https://github.com/davedleo/FedGWC

## 1. Introduction

Federated Learning (FL) (McMahan et al., 2017) has emerged as a promising paradigm for training models on decentralized data while preserving privacy. Unlike tradi-

---
[*]Equal contribution [1]Department of Mathematical Sciences, Polytechnic University of Turin, Italy [2]Istituto Nazionale di Fisica Nucleare (INFN), Sezione di Torino, Turin, Italy [3]Department of Computing and Control Engineering, Polytechnic University of Turin, Italy [4]Basque Center for Applied Mathematics (BCAM), Bilbao, Spain [5]Vector Institute, Toronto, Ontario, Canada. [†] Work started when the author was at the Polytechnic University of Turin. Correspondence to: Alessandro Licciardi <alessandro.licciardi@polito.it>.

*Proceedings of the 42$^{nd}$ International Conference on Machine Learning*, Vancouver, Canada. PMLR 267, 2025. Copyright 2025 by the author(s).

tional machine learning frameworks, FL enables collaborative training between multiple clients without requiring data transfer, making it particularly attractive in privacy-sensitive domains (Bonawitz et al., 2019). FL was introduced primarily to address two major challenges in decentralized scenarios: ensuring privacy (Kairouz et al., 2021) and reducing communication overhead (Hamer et al., 2020; Asad et al., 2020). In particular, FL algorithms must guarantee *communication efficiency*, to reduce the burden associated with the exchange of model updates between clients and the central server, while maintaining *privacy*, as clients should not expose their private data during the training process.

A core challenge in federated learning is data heterogeneity (Li et al., 2020). This manifests in two key ways: through imbalances in data quantity and class distribution both within and across clients, and through the non-independent and non-identical distribution (non-IID) of data across the federation. In this scenario, a single global model often fails to generalize due to clients contributing with updates from skewed distributions, leading to degraded performance (Zhao et al., 2018; Caldarola et al., 2022) compared to the centralized counterpart. Furthermore, noisy or corrupted data from some clients can further complicate the learning process (Cao et al., 2020; Zhang et al., 2022), while non-IID data often results in unstable convergence and conflicting gradient updates (Hsieh et al., 2020; Zhao et al., 2018). Despite the introduction of various techniques to mitigate these issues – ranging from regularization methods (Li et al., 2020) and optimizer modifications based on momentum (Mendieta et al., 2022) or control variates (Karimireddy et al., 2020b), to model-level strategies such as feature distillation (Yang et al., 2023), and deeper investigations into solution landscape properties like achieving flatter minima (Lee & Yoon, 2024; Fan et al., 2024) or preventing dimensional collapse (Shi et al., 2023) – data heterogeneity persists as a critical problem.

In this work, we address the fundamental challenges of data heterogeneity and class imbalance in federated learning through a novel clustering-based approach. We propose `FedGWC` (Federated Gaussian Weighting Clustering), a method that groups clients with similar data distributions into clusters, enabling the training of personalized federated models for each group. Our key insight is that clients' data distributions can be inferred by analyzing their empir-

ical loss functions, rather than relying on model updates as most existing approaches do. Inspired by (Cho et al., 2022), we hypothesize that clients with similar data distributions will exhibit similar loss landscapes. `FedGWC` implements this insight using a Gaussian reward mechanism to form homogeneous clusters based on an *interaction matrix*, which encodes the pairwise similarity of clients' data distributions. This clustering is achieved efficiently by having clients communicate only their empirical losses to the server at each communication round. Our method transforms these loss values to estimate the similarity between each client's data distribution and the global distribution, using Gaussian weights as statistical estimators. Each cluster then trains its own specialized federated model, leveraging the shared data characteristics within that group. This approach preserves the knowledge-sharing benefits of federated learning while reducing the negative effects of statistical heterogeneity, such as client drift (Karimireddy et al., 2020b). We develop a comprehensive mathematical framework for this approach and rigorously prove the convergence properties of the Gaussian weights estimators.

To evaluate our method, we introduce the *Wasserstein Adjusted Score*, a new clustering metric tailored for assessing cluster cohesion in class-imbalanced FL scenarios. Through extensive experiments on both benchmark (Caldas et al., 2018) and large-scale datasets (Hsu et al., 2020), we demonstrate that `FedGWC` outperforms existing clustered FL algorithms in terms of both accuracy and clustering quality. Furthermore, `FedGWC` can be integrated with any robust FL aggregation algorithm to provide additional resilience against data heterogeneity.

**Contributions.**

- We propose `FedGWC`, an efficient federated learning framework that clusters clients based on their data distributions, enabling personalized models that better handle heterogeneity.
- We provide a rigorous mathematical framework to motivate the algorithm, proving its convergence properties and providing theoretical guarantees for our clustering approach.
- We introduce a novel clustering metric specifically designed to evaluate cluster quality in the presence of class imbalance.
- We demonstrate through extensive experiments that 1) `FedGWC` outperforms existing clustered FL approaches in terms of both clustering quality and model performance, 2) our method successfully handles both class and domain imbalance scenarios, and 3) the framework can be effectively integrated with any FL aggregation algorithm.

## 2. Related Work

**FL with Heterogeneous Data.** Handling data heterogeneity, especially class imbalance, remains a critical challenge in FL. `FedProx`(Li et al., 2020) was one of the first attempts to address heterogeneity by introducing a proximal term that constrains local model updates close to the global model. `FedMD` (Li & Wang, 2019) focuses on heterogeneity in model architectures, allowing collaborative training between clients with different neural network structures using model distillation. Methods such as `SCAFFOLD` (Karimireddy et al., 2020b) and `Mime` (Karimireddy et al., 2020a) have also been proposed to reduce client drift by using control variates during the optimization process, which helps mitigate the effects of non-IID data. Furthermore, strategies such as biased client selection (Cho et al., 2022) based on ranking local losses of clients and normalization of updates in `FedNova` (Wang et al., 2021) have been developed to specifically address class imbalance in federated networks, leading to more equitable global model performance.

**Clustered FL.** Clustering has proven to be an effective strategy in FL for handling client heterogeneity and improving personalization (Huang et al., 2022; Duan et al., 2021; Briggs et al., 2020; Caldarola et al., 2021; Ye et al., 2023). Clustered FL (Sattler et al., 2020) is one of the first methods proposed to group clients with similar data distributions to train specialized models rather than relying on a single global one. Nevertheless, from a practical perspective, this method exhibits pronounced sensitivity to hyper-parameter tuning, especially concerning the gradient norms threshold, which is intricately linked to the dataset. This sensitivity can result in significant issues of either excessive under-splitting or over-splitting. Additionally, as client sampling is independent of the clustering, there may be privacy concerns due to the potential for updating cluster models with the gradient of a single client. An extension of this is the efficient framework for clustered FL proposed by (Ghosh et al., 2020), which strikes a balance between model accuracy and communication efficiency. Multi-Center FL (Long et al., 2023) builds on this concept by dynamically adjusting client clusters to achieve better personalization, however a-priori knowledge on the number of clusters is needed. Similarly, `IFCA` (Ghosh et al., 2020) addresses client heterogeneity by predefining a fixed number of clusters and alternately estimating the cluster identities of the users by optimizing model parameters for the user clusters via gradient descent. However, it imposes a significant computational burden, as the server communicates all cluster models to each client, which must evaluate every model locally to select the best fit based on loss minimization. This approach not only increases communication overhead but also introduces inefficiencies, as each client must test all models, making it less scalable in larger networks. Recent studies on clustered FL investigate sophisticated client

grouping strategies. For instance, (Carrillo et al., 2024) employs consensus-based optimization dynamics, where iterative interactions and calculations across all client representations, viewed as particles, are typically required to guide them towards a group consensus on cluster membership; this process can be computationally demanding, especially as the number of clients and the complexity of their models increase. Another approach (Bao et al., 2023) involves training pairwise discriminators between all, or a significant number of, potential client pairs to estimate their data distribution similarities. This approach may result in a quadratic increase in the number of auxiliary models for the exclusive aim of clustering, thus imposing a substantial computational burden, particularly within federated networks with a vast number of clients.

Compared to previous approaches, the key advantage of the proposed algorithm, FedGWC, lies in its ability to effectively identify clusters of clients with similar levels of heterogeneity and class distribution through simple transformations of the individual empirical loss process. This is achieved without imposing significant communication overhead or requiring additional computational resources. Further details on the computational and communication overhead are provided in Appendix D. Additionally, FedGWC can be seamlessly integrated with any aggregation method, enhancing its robustness and performance when dealing with heterogeneous scenarios.

## 3. Problem Formulation

Consider a standard FL scenario (McMahan et al., 2017) with $K$ clients and a central server. FL typically addresses the following optimization problem $\min_{\theta \in \Theta} \mathcal{L}(\theta) = \min_{\theta \in \Theta} \sum_{k=1}^{K} \frac{n_k}{n} \mathcal{L}_k(\theta)$, where $\mathcal{L}_k(\cdot)$ represents the loss function of client $k$, $n_k$ is the number of training samples on client $k$, $n = \sum_{k=1}^{K} n_k$ is the total number of samples, and $\Theta$ denotes the model's parameters space. At each communication round $t \in [T]$, a subset $\mathcal{P}_t$ of clients is selected to participate in training. Each participating client performs $S$ iterations, updating its local parameters using a stochastic optimizer, e.g. Stochastic Gradient Descent (SGD). In clustered FL, the objective is to partition clients into non-overlapping groups $\mathcal{C}^{(1)}, \ldots, \mathcal{C}^{(n_{cl})}$ based on similarities in their data distributions, with each group having its own model, $\theta_{(1)}, \ldots, \theta_{(n_{cl})}$.

## 4. FedGWC Algorithm

In this section, we present the core components of FedGWC in a progressive manner. First, Subsection 4.1 introduces the Gaussian Weighting mechanism, a statistical method that estimates how well each client's data distribution aligns with the overall federation. Subsection 4.2 then explains how we model and detect clusters by analyzing interactions

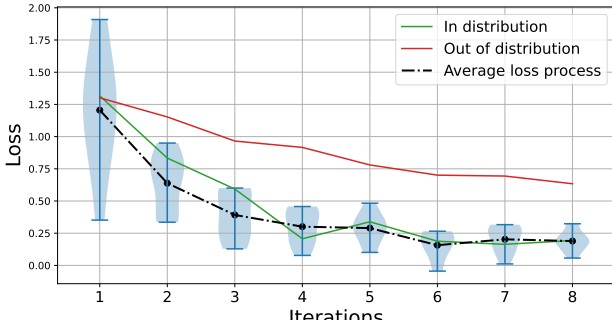

*Figure 1.* Illustration of the Gaussian reward mechanism for two clients from Cifar100 (Dirichlet $\alpha = 0.05$, 10 sampled clients per round and $S = 8$ local iterations). The dashed line represents the average loss process $m^{t,s}$, with the blue region indicating the confidence interval $m^{t,s} \pm \sigma^{t,s}$ at fixed $t$, $s = 1, \ldots, 8$. The green curve corresponds to an in-distribution client, whose loss remains within the confidence region, resulting in a high Gaussian reward. The red line represents an out-of-distribution client, whose loss lies outside the confidence region, resulting in a lower reward.

between client distributions. For clarity, we first present these concepts considering the federation as a single cluster, focusing on the fundamental mechanisms that enable client grouping. Finally, Subsection 4.3 introduces the complete algorithmic framework, including cluster indices and the iterative structure that allows FedGWC to partition clients into increasingly homogeneous groups.

### 4.1. Gaussian Weighting Mechanism

To assess how closely the local data of each client aligns with the global distribution, we introduce the *Gaussian Weights* $\gamma_k$, statistical estimators that capture the closeness of each clients' distribution to the main distribution of the cluster. A weight near zero suggests that the client's distribution is far from the main distribution. We graphically represent the idea of the Gaussian rewards in Figure 1.

The fundamental principle of FedGWC is to group clients based on the similarity of their empirical losses, which are used to assign a *reward* between 0 and 1 to each client at each local iteration. A high reward indicates that a client's loss is close to the cluster's mean loss, while a lower reward reflects greater divergence. Gaussian weights estimate the expected value of these rewards, quantifying the closeness between each client's distribution and the global one.

Every communication round $t$, each sampled client $k \in \mathcal{P}_t$ communicates the server the empirical loss process $l_k^{t,s} = \mathcal{L}(\theta_k^{t,s})$, for $s = 1, \ldots, S$, alongside the updated model $\theta_k^{t+1}$. The server computes the *rewards* as

$$r_k^{t,s} = \exp\left(-\frac{(l_k^{t,s} - m^{t,s})^2}{2(\sigma^{t,s})^2}\right) \in (0,1) \quad, \quad (1)$$

where $m^{t,s} = 1/|\mathcal{P}_t| \sum_{k \in \mathcal{P}_t} l_k^{t,s}$ is the average loss process w.r.t. the local iterations $s = 1, \ldots, S$, and $(\sigma^{t,s})^2 =$

$1/(|\mathcal{P}_t| - 1) \sum_{k \in \mathcal{P}_t} (l_k^{t,s} - m^{t,s})^2$ is the sample variance. The rewards $r_k^{t,s} \to 1$ as the distance between $l_k^{t,s}$ and the average process $m^{t,s}$ decreases, while $r_k^{t,s} \to 0$ as the distance increases. In Eq. 1, during any local iteration $s$, a Gaussian kernel centered on the mean loss and with spread the sample variance assesses the clients' proximity to the confidence interval's center, indicating their probability of sharing the same learning process distribution. Since the values of the rewards can suffer from stochastic oscillations, we reduce the noise in the estimate by averaging the rewards over the $S$ local iterations, thus obtaining $\omega_k^t = 1/S \sum_{s \in [S]} r_k^{t,s}$ for each sampled client $k$. Instead of using a single sample, such as the last value of the loss as done in (Cho et al., 2022), we opted for averaging across iterations to provide a more stable estimate. However, the averaged rewards $\omega_k^t$ depend on the sampled set of clients $\mathcal{P}_t$ and on the current round. Hence, we introduce the **Gaussian weights** $\gamma_k^t$ to keep track over time of these rewards. The Gaussian weights are computed via a running average of the instant rewards $\omega_k^t$. In particular, for each client $k$ the weight $\gamma_k^0$ is initialized to 0 to avoid biases in the expectation estimate, and when it is randomly selected is update according to

$$\gamma_k^{t+1} = (1 - \alpha_t)\gamma_k^t + \alpha_t \omega_k^t \qquad (2)$$

for a sequence of coefficients $\alpha_t \in (0,1)$ for any $t \in [T]$. The weight definition in Eq.2 is closely related to the Robbins-Monro stochastic approximation method (Robbins & Monro, 1951). If a client is not participating in the training, its weight is not updated. FedGWC mitigates biases in the estimation of rewards by employing two mechanisms: (1) uniform random sampling method for clients, with a dynamic adjustment process to prioritize clients that are infrequently sampled, thus ensuring equitable participation across time periods; and (2) when a client is not sampled in a round, its weight and contribution to the reward estimate remain unchanged.

### 4.2. Modeling Interactions with Gaussian Weights

**Interaction Matrix.** Gaussian weights are scalar quantities that offer an absolute measure of the alignment between a client's data distribution and the global distribution. Although these weights indicate the conformity of each client's distribution individually, they do not consider the interrelations among the distributions of different clients. Therefore, we propose to encode these interactions in an *interaction matrix* $P^t \in \mathbb{R}^{K \times K}$ whose element $P_{kj}^t$ estimates the similarity between the $k$-th and the $j$-th client data distribution. The interaction matrix is initialized to the null matrix, *i.e.* $P_{kj}^0 = 0$ for every couple $k, j \in [K]$.

Specifically, we define the update rule for the matrix $P^t$ as follows:

$$P_{kj}^{t+1} = \begin{cases} (1 - \alpha_t)P_{kj}^t + \alpha_t \omega_k^t, & (k,j) \in \mathcal{P}_t \times \mathcal{P}_t \\ P_{kj}^t, & (k,j) \notin \mathcal{P}_t \times \mathcal{P}_t \end{cases} \qquad (3)$$

where $\{\alpha_t\}_t$ is the same sequence used to update the weights, and $\mathcal{P}_t$ is the subset of clients sampled in round $t$.

Intuitively, in the long run, since $\omega_k^t$ measures the proximity of the loss process of client $k$ to the average loss process of clients in $\mathcal{P}_t$ at round $t$, we are estimating the *expected perception* of client $k$ by client $j$ with $P_{kj}^t$, *i.e.* a larger value indicates a higher degree of similarity between the loss profiles, whereas smaller values indicate a lower degree of similarity. For example, if $P_{kj}^t$ is close to 1, it suggests that on average, whenever $k$ and $j$ have been simultaneously sampled prior to round $t$, $\omega_k^t$ was high, meaning that the two clients are well-represented within the same distribution.

To effectively extract the information embedded in $P$, we introduce the concept of *unbiased perception vectors* (UPV). For any pair of clients $k, j \in [K]$, the UPV $v_k^j \in \mathbb{R}^{K-2}$ represents the $k$-th row of $P$, excluding the $k$-th and $j$-th entries. Recalling the construction of $P^t$, where each row indicates how a client is perceived to share the same distribution as other clients in the federation, the UPV $v_k^j$ captures the collective perception of client $k$ by all other clients, excluding both itself and client $j$. This exclusion is why we refer to $v_k^j$ as *unbiased*.

The UPVs encode information about the relationships between clients, which can be exploited for clustering. However, the UPVs cannot be directly used as their entries are only aligned when considered in pairs. Instead, we construct the *affinity matrix* $W$ by transforming the information encoded by the UPVs through an RBF kernel, as this choice allows to effectively model the affinity between clients: two clients are considered *affine* if similarly perceived by others. This relation is encoded by the entries of $W \in \mathbb{R}^{K \times K}$, which we define as:

$$W_{kj} = \mathcal{K}(v_k^j, v_j^k) = \exp\left(-\beta \left\| v_k^j - v_j^k \right\|^2 \right). \qquad (4)$$

The spread of the RBF kernel is controlled by a single hyperparameter $\beta > 0$: changes in this value provide different clustering outcomes, as shown in the sensitivity analysis in Appendix G.

**Clustering.** The affinity matrix $W$, designed to be symmetric, highlights features that capture similarities between clients' distributions. Clustering is performed by the server using the rows of $W$ as feature vectors, as they contain the relevant information. We apply the spectral clustering algorithm (Ng et al., 2001) to $W$ due to its effectiveness in detecting non-convex relationships embedded within the client affinities. Symmetrizing the interaction matrix $P$ into the affinity matrix $W$ is fundamental for spectral clustering as it refines inter-client relationship representation. It models interactions, reducing biases, and emphasizing reliable similarities. This improves robustness to noise, allowing spectral

clustering to effectively detect the distributional structure underlying the clients' network (Von Luxburg, 2007). During the iterative training process, the server determines whether to perform clustering by checking the convergence of the matrix $P^t$. Convergence is numerically verified when the mean squared error (MSE) between consecutive updates is less than a small threshold $\epsilon > 0$. Algorithm 2 summarizes the clustering procedure. If the MSE is below $\epsilon$, the server computes the matrix $W^t$ and performs spectral clustering over $W^t$ with a number of clusters $n \in \{2, \ldots, n_{max}\}$. For each clustering outcome, the Davies-Bouldin (DB) score (Davies & Bouldin, 1979) is computed: DB larger than one means that clusters are not well separated, while if it is smaller than one, the clusters are well separated, a detailed description of the clustering metrics is provided in Appendix B. We denote by $n_{cl}$ the optimal number of clusters detected by FedGWC. If $\min_{n=2,\ldots,n_{max}} DB_n > 1$, we do not split the current cluster. Hence, the optimal number of clusters is $n_{cl}$ is one. In the other case, the optimal number of clusters is $n_{cl} \in \arg\min_{n=2,\ldots,n_{max}} DB_n$. This requirement ensures proper control over the over-splitting phenomenon, a common issue in hierarchical clustering algorithms in FL which can undermine key principles of FL by creating degenerate clusters with very few clients. Finally, on each cluster $\mathcal{C}^{(1)}, \ldots, \mathcal{C}^{(n_{cl})}$ an FL aggregation algorithm is trained separately, resulting in models $\theta_{(1)}, \ldots, \theta_{(n_{cl})}$ personalized for each cluster.

### 4.3. FedGWC Implementation

In the previous sections, we have detailed the splitting algorithm within the individual clusters. In this section, we present the full FedGWC procedure (Algorithm 1), introducing the complete notation with indices for the distinct clusters. We denote the clustering index as $n$, and the total number of clusters $N_{cl}$.

The interaction matrix $P_{(1)}^0$ is initialized to the null matrix $0_{K \times K}$, and the *total number of clusters* $N_{cl}^0$, as no clusters have been formed yet, and $\text{MSE}_{(1)}^0$ are initialized to 1, in order to ensure stability in early updates, allowing a gradual decrease. At each communication round $t$, and for cluster $\mathcal{C}^{(n)}$, where $n = 1, \ldots, N_{cl}^t$, the cluster server independently samples the participating clients $\mathcal{P}_t^{(n)} \subseteq \mathcal{C}^{(n)}$. Each client $k \in \mathcal{P}_t^{(n)}$ receives the current cluster model $\theta_{(n)}^t$. After performing local updates, each client sends its updated model $\theta_k^{t+1}$ and empirical loss $l_k^t$ back to the cluster server. The server aggregates these updates to form the new cluster model $\theta_{(n)}^{t+1}$, computes the Gaussian rewards $\omega_k^t$ for the sampled clients, and updates the interaction matrix $P_{(n)}^{t+1}$ and $\text{MSE}_{(n)}^{t+1}$ according to Eq. 3. If $\text{MSE}_{(n)}^{t+1}$ is lower than a threshold $\epsilon$, the server of the cluster performs clustering to determine whether to split cluster $\mathcal{C}^{(n)}$ into $n_{cl}$ sub-groups, as outlined in Algorithm 2. The matrix $P_{(n)}^{t+1}$ is then parti-

---

**Algorithm 1** FedGWC

1: **Input:** $K, T, S, \alpha_t, \epsilon, |\mathcal{P}_t|, \mathcal{K}$
2: **Output:** $\mathcal{C}^{(1)}, \ldots, \mathcal{C}^{(N_{cl})}$ and $\theta_{(1)}, \ldots, \theta_{(N_{cl})}$
3: Initialize $N_{cl}^0 \leftarrow 1$
4: Initialize $P_{(1)}^0 \leftarrow 0_{K \times K}$
5: Initialize $\text{MSE}_{(1)}^0 \leftarrow 1$
6: **for** $t = 0, \ldots, T - 1$ **do**
7:    $\Delta N^t \leftarrow 0$ for each iterations it counts the number of new clusters that are detected
8:    **for** $n = 1, \ldots, N_{cl}^t$ **do**
9:       Server samples $\mathcal{P}_t^{(n)} \in \mathcal{C}^{(n)}$ and sends the current cluster model $\theta_{(n)}^t$
10:       Each client $k \in \mathcal{P}_t^{(n)}$ locally updates $\theta_k^t$ and $l_k^t$, then sends them to the server
11:       $\omega_k^t \leftarrow$ Gaussian_Rewards$(l_k^t, \mathcal{P}_t^{(n)})$, Eq. 1
12:       $\theta_{(n)}^{t+1} \leftarrow$ FL_Aggregator$(\theta_k^t, \mathcal{P}_t^{(n)})$
13:       $P_{(n)}^{t+1} \leftarrow$ Update_Matrix$(P_{(n)}^t, \omega_k^t, \alpha_t, \mathcal{P}_t^{(n)})$, according to Eq. 3
14:       Update $\text{MSE}_{(n)}^{t+1}$
15:       **if** $\text{MSE}_{(n)}^{t+1} < \epsilon$ **then**
16:          Perform    FedGW_Cluster$(P_{(n)}^{t+1}, n_{max}, \mathcal{K})$ on $\mathcal{C}^{(n)}$, providing $n_{cl}$ sub-clusters
17:          Update the number of new clusters $\Delta N^t \leftarrow \Delta N^t + n_{cl} - 1$
18:          Cluster server splits $P_{(n)}^{t+1}$ filtering rows and columns according to the new clusters
19:          Re-initialize MSE for new clusters to 1
20:       **end if**
21:    **end for**
22:    Update the total number of clusters $N_{cl}^{t+1} \leftarrow N_{cl}^t + \Delta N^t$
23: **end for**

---

tioned into sub-matrices by filtering its columns and rows according to the newly formed clusters, with the MSE for these sub-matrices reinitialized to 1. This process results in a distinct model $\theta_{(n)}$ for each cluster $\mathcal{C}^{(n)}$. When the final iteration $T$ is reached we are left with $N_{cl}^T$ clusters with personalized models $\theta_{(n)}$ for $n = 1, \ldots, N_{cl}^T$.

Thanks to the Gaussian Weights, and the iterative spectral clustering on the affinity matrices, our algorithm, FedGWC, is able to autonomously detect groups of clients that display similar levels of heterogeneity. The clusters formed are more uniform, *i.e.* the class distributions within each group are more similar. These results are supported by experimental evaluations, discussed in Section 7.3.

# 5. Theoretical Results and Derivation of `FedGWC`

In this section, we provide a formal derivation of the algorithm, discussing the mathematical properties of Gaussian Weights and outlining the structured formalism and rationale underlying `FedGWC`.

The stochastic process induced by the optimization algorithm in the local update step, allows the evolution of the empirical loss to be modeled using random variables within a probabilistic framework. We denote random variables with capital letters (*e.g.*, $X$), and their realizations with lowercase letters (*e.g.*, $x$).

The observed loss process $l_k^{t,s}$ is the outcome of a stochastic process $L_k^{t,s}$, and the rewards $r_k^{t,s}$, computed according to Eq. 1, are samples from a random reward $R_k^{t,s}$ supported in $(0, 1)$, whose expectation $\mathbb{E}[R_k^{t,s}]$ is lower for out-of-distribution clients and higher for in-distribution ones. To estimate the expected reward $\mathbb{E}[R_k^{t,s}]$ we introduce the r.v. $\Omega_k^t = 1/S \sum_{s \in [S]} R_k^{t,s}$, which is an estimator less affected by noisy fluctuations in the empirical loss. Due to the linearity of the expectation operator, the expected reward $\mathbb{E}[R_k^{t,s}]$ for the $k$-th client at round $t$, local iteration $s$ equals the expected Gaussian reward $\mathbb{E}[\Omega_k^t]$ that, to simplify the notation, we denote by $\mu_k$. $\mu_k$ is the theoretical value that we aim to estimate by designing our Gaussian weights $\Gamma_k^t$ appropriately, as it encodes the ideal reward to quantify the closeness of the distribution of each client $k$ to the main distribution. Note that the process is stationary by construction. Therefore, it does not depend on $t$ but differs between clients, as it reaches a higher value for in-distribution clients and a lower for out-of-distribution clients.

To rigorously motivate the construction of our algorithm and the reliability of the weights, we introduce the following theoretical results. Theorems 5.1 and 5.2 demonstrate that the weights converge to a finite value and, more importantly, that this limit serves as an unbiased estimator of the theoretical reward $\mu_k$. The first theorem provides a strong convergence result, showing that, with suitable choices of the sequence $\{\alpha_t\}_t$, the expectation of the Gaussian weights $\Gamma_k^t$ converges to $\mu_k$ in $L^2$ and almost surely. In addition, Theorem 5.2 extends this to the case of constant $\alpha_t$, proving that the weights still converge and remain unbiased estimators of the rewards as $t \to \infty$.

**Theorem 5.1.** *Let $\{\alpha_t\}_{t=1}^\infty$ be a sequence of positive real values, and $\{\Gamma_k^t\}_{t=1}^\infty$ the sequence of Gaussian weights. If $\{\alpha_t\}_{t=1}^\infty \in l^2(\mathbb{N})/l^1(\mathbb{N})$, then $\Gamma_k^t$ converges in $L^2$. Furthermore, for $t \to \infty$,*

$$\Gamma_k^t \longrightarrow \mu_k \quad a.s. \tag{5}$$

**Theorem 5.2.** *Let $\alpha \in (0, 1)$ be a fixed constant, then in the limit $t \to \infty$, the expectation of the weights converges to the individual theoretical reward $\mu_k$, for each client $k = 1, \ldots, K$, i.e.,*

$$\mathbb{E}[\Gamma_k^t] \longrightarrow \mu_k \quad t \to \infty. \tag{6}$$

Proposition 5.3 shows that Gaussian weights reduce the variance of the estimate, thus decreasing the error and enabling the construction of a confidence interval for $\mu_k$.

**Proposition 5.3.** *The variance of the weights $\Gamma_k^t$ is smaller than the variance $\sigma_k^2$ of the theoretical rewards $R_k^{t,s}$.*

Complete proofs of Theorems 5.1, 5.2, and Proposition 5.3 are provided in Appendix A. Additionally, Appendix A includes further analysis of `FedGWC`. Specifically, Proposition A.4 demonstrates that the entries of the interaction matrix $P$ are bounded, while Theorem A.5 establishes a sufficient condition for conserving the sampling rate during the recursive procedure.

# 6. Wasserstein Adjusted Score

In the previous Section we observed that when clustering clients according to different heterogeneity levels, the outcome must be evaluated using a metric that assesses the cohesion of individual distributions. In this Section, we introduce a novel metric to evaluate the performance of clustering algorithms in FL. This metric, derived from the Wasserstein distance (Kantorovich, 1942), quantifies the cohesion of client groups based on their class distribution similarities.

We propose a general method for adapting clustering metrics to account for class imbalances. This adjustment is particularly relevant when the underlying class distributions across clients are skewed. The formal derivation and mathematical details of the proposed metric are provided in Appendix B. We now provide a high-level overview of our new metric.

Consider a generic clustering metric $s$, e.g. Davies-Bouldin score (Davies & Bouldin, 1979) or the Silhouette score (Rousseeuw, 1987). Let $C$ denote the total number of classes, and $x_i^k$ the empirical frequency of the $i$-th class in the $k$-th client's local training set. Following theoretical reasonings, as shown in Appendix B, the empirical frequency vector for client $k$, denoted by $x_{(i)}^k$, is ordered according to the rank statistic of the class frequencies, *i.e.* $x_{(i)}^k \geq x_{(i+1)}^k$ for any $i = 1, \ldots, C - 1$. The class-adjusted clustering metric $\tilde{s}$ is defined as the standard clustering metric $s$ computed on the ranked frequency vectors $x_{(i)}^k$. Specifically, the distance between two clients $j$ and $k$ results in

$$\frac{1}{C} \left( \sum_{i=1}^{C} \left| x_{(i)}^k - x_{(i)}^j \right|^2 \right)^{1/2}. \tag{7}$$

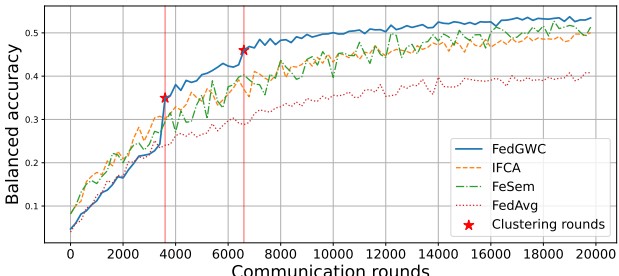

*Figure 2.* Balanced accuracy on Cifar100 for `FedGWC` (blue curve) with `FedAvg` aggregation compared to the clustered FL baselines. `FedGWC` detects two splits demonstrating significant improvements in accuracy when clustering is performed, leading also to a faster and more stable convergence than baseline algorithms.

This modification ensures that the clustering evaluation is sensitive to the distributional characteristics of the class imbalance. As we show in Appendix B, this adjustment is mathematically equivalent to assessing the dispersion between the empirical class probability distributions of different clients using the Wasserstein distance, also known as the Kantorovich-Rubenstein metric (Kantorovich, 1942). This equivalence highlights the theoretical soundness of using ranked class frequencies to better capture variations in class distributions when evaluating clustering outcomes in FL.

## 7. Experiments

In this section, we present the experimental results on widely used FL benchmark datasets (Caldas et al., 2018) including real-world datasets (Hsu et al., 2020), comparing the performance of `FedGWC` with other baselines from the literature, including standard FL algorithms and clustering methods. A detailed description of the implementation settings, datasets and models used for the evaluation are reported in Appendix F. In Section 7.1, we evaluate our method, `FedGWC`, against various clustering algorithms, including `CFL` (Sattler et al., 2020), `FeSEM` (Long et al., 2023), and `IFCA` (Ghosh et al., 2020), and standard FL aggregations `FedAvg` (McMahan et al., 2017), `FedAvgM` (Asad et al., 2020), FairAvg (Michieli & Ozay, 2021) and `FedProx` (Li et al., 2020), showing also that how our approach is orthogonal to conventional FL aggregation methods.

In Section 7.2, we underscore that `FedGWC` has the capability to surpass FL methods in real-world and large-scale scenarios (Hsu et al., 2020).

Finally, in Section 7.3, we propose analyses on class and domain imbalance, showing that our algorithm successfully detects clients belonging to separate distributions. Further experiments are presented in Appendix J. Each client has its own local train and test sets.

*Table 1.* FL baselines in heterogeneous scenarios. Clustering baselines use FedAvg as aggregation mechanism. We emphasize the fact that `FedGWC` and `CFL` automatically detect the number of clusters, unlike `IFCA` and `FeSEM` which require tuning the number of clusters. A higher WAS , denoted by ↑, and a lower WADB, denoted by ↓ indicate better clustering outcomes

| | FL method | C | Automatic Cluster Selection | Acc | WAS ↑ | WADB ↓ |
|---|---|---|---|---|---|---|
| Cifar100 Clustered FL | IFCA | 5 | ✗ | $47.5 \pm 3.5$ | $-0.8 \pm 0.2$ | $5.2 \pm 5.1$ |
| | FeSem | 5 | ✗ | $53.4 \pm 1.8$ | $-0.3 \pm 0.1$ | $38.4 \pm 13.0$ |
| | CFL | 1 | ✓ | $41.6 \pm 1.3$ | / | / |
| | FedGWC | 4 | ✓ | $\mathbf{53.4 \pm 0.4}$ | $\mathbf{0.1 \pm 0.0}$ | $\mathbf{2.4 \pm 0.4}$ |
| Cifar100 Classic FL | FedAvg | 1 | / | $41.6 \pm 1.3$ | / | / |
| | FedAvgM | 1 | / | $41.5 \pm 0.5$ | / | / |
| | FedProx | 1 | / | $41.8 \pm 1.0$ | / | / |
| Femnist Clustered FL | IFCA | 5 | ✗ | $76.7 \pm 0.6$ | $\mathbf{0.3 \pm 0.1}$ | $\mathbf{0.5 \pm 0.1}$ |
| | FeSem | 2 | ✗ | $75.6 \pm 0.2$ | $0.0 \pm 0.0$ | $25.6 \pm 7.8$ |
| | CFL | 1 | ✓ | $76.0 \pm 0.1$ | / | / |
| | FedGWC | 4 | ✓ | $76.1 \pm 0.1$ | $-0.2 \pm 0.1$ | $18.0 \pm 6.2$ |
| Femnist Classic FL | FedAvg | 1 | / | $76.6 \pm 0.1$ | / | / |
| | FedAvgM | 1 | / | $\mathbf{83.3 \pm 0.3}$ | / | / |
| | FedProx | 1 | / | $75.9 \pm 0.2$ | / | / |

Algorithm performance is evaluated by averaging the accuracy achieved on the local test sets across the federation, enabling a comparison between FL aggregation and clustered FL approaches (refer to Appendix F for additional insights). When assessing clustering baselines, we also use the Wasserstein's Adjusted Silhouette Score (WAS) and Wasserstein's Adjusted Davies-Bouldin Score (WADB) to quantify the distributional cohesion among clients, an evaluation performed *a posteriori*. For detection tasks in visual domains (Section 7.3), we compute the Rand Index (Rand, 1971), a clustering metric that compares the obtained clustering with a ground truth labeling. Further details on the chosen metrics are provided in Appendices E and B.

### 7.1. `FedGWC` in heterogeneous settings

In this section, we analyze the effectiveness of `FedGWC` in mitigating the impact of data heterogeneity compared to standard aggregation methods and other clustered FL algorithms. We conduct experiments on Cifar100 (Krizhevsky et al., 2009) with 100 clients and Femnist (LeCun, 1998) with 400 clients, controlling heterogeneity through a Dirichlet parameter $\alpha$, set to 0.5 for Cifar100 and 0.01 for Femnist, reflecting a realistic class imbalance across clients. Implementation details are provided in Appendix F.

We compare `FedGWC` against clustered FL baselines (`IFCA`, `FeSEM`, `CFL`) using `FedAvg` aggregation, as well as standard FL algorithms (`FedAvg`, `FedAvgM`, `FedProx`). For algorithms requiring a predefined number of clusters (`IFCA`, `FeSEM`), we report the best result among 2, 3, 4, and 5 clusters, with full tuning details in Appendix I. While `IFCA` achieves competitive results, its high communication overhead—requiring each client to evaluate models from every cluster in each round—makes it impractical for cross-device

FL, serving as an upper bound in our study. `FeSEM` is more efficient than `IFCA` but lacks adaptability due to its fixed cluster count. Meanwhile, `CFL` requires extensive hyper-parameter tuning and often produces overly fine-grained clusters or fails to form clusters altogether. In contrast, `FedGWC` requires only one hyperparameter and provides a more practical clustering strategy for cross-device FL.

Table 1 presents a comparative analysis of these algorithms in terms of balanced accuracy, WAS, and WADB, using `FedAvg` as the aggregation method. Higher WAS values indicate better clustering, while lower WADB values suggest better cohesion. On Femnist, clustering-based methods perform worse than standard FL aggregation, but as we move to the more complex and realistic Cifar100 scenario, it becomes evident that clustered FL is necessary to address heterogeneity. In this case, `FedGWC` achieves the best performance in both classification accuracy and clustering quality, with the latter directly influencing the former. The need for clustering grows with increasing heterogeneity, as seen in Table 1: standard FL approaches struggle when trained on a single heterogeneous cluster, whereas clustered FL effectively mitigates the heterogeneity effect. This is particularly relevant for Cifar100, which has a larger number of classes and three-channel images, whereas Femnist consists of grayscale images from only 47 classes.

In Table 1, we present a comparative analysis of these algorithms with respect to balanced accuracy, WAS, and WADB, employing `FedAvg` as the aggregation strategy. Recall that higher the value of WAS the better the clustering outcome, as, for WADB, a lower value suggests a better cohesion between clusters. Further details on the metrics used are provided in Appendix E.

Notably, both `FedGWC` and `CFL` automatically determine the optimal number of clusters based on data heterogeneity, offering a more scalable solution for large-scale cross-device FL. In contrast to `CFL`, `FedGWC` consistently produced a reasonable number of clusters, even when using the optimal hyperparameters for `CFL`, which resulted in no splits, thereby achieving performance equivalent to FedAvg.

We observe in Figure 2 that `FedGWC` exhibits a significant improvement in accuracy on Cifar100 precisely at the rounds where clustering occurs.

As detailed in Table 9 in Appendix J, `FedGWC` is orthogonal to any FL aggregation algorithm, *i.e.* any FL method can be easily embedded in `FedGWC`. Our method consistently boosted the performance of FL algorithms for the more heterogeneous settings of Cifar100 and Femnist.

### 7.2. `FedGWC` in Large Scale and Real World Scenarios

We evaluate `FedGWC` on two large-scale, real-world datasets: Google Landmarks (Weyand et al., 2020) and iNat-

uralist (Van Horn et al., 2018), respectively considering the partitions Landmarks-Users-160K, and iNaturalist-Users-120K, proposed in (Hsu et al., 2020). Both datasets exhibit high data heterogeneity and involve a large number of clients - approximately 800 for Landmarks and 2,700 for iNaturalist. To simulate a realistic cross-device scenario, we set 10 participating clients per round. For this experiment, we compare `FedGWC` against clustered FL baselines (`IFCA`, `CFL`) and standard FL aggregation methods (`FedAvg`, `FedAvgM`, `FedProx`, `FairAvg`). The number of clusters for `IFCA` is tuned between 2 and 5. Due to its high computational cost in large-scale settings, `FeSEM` was not included in this analysis. Table 2 reports the results: `FedGWC` with `FedAvg` aggregation achieves 57.4% accuracy on Landmarks, significantly outperforming all standard FL methods. In this scenario, `FedGWC` detects 5 clusters with the best hyperparameter setting ($\beta = 0.5$), while `IFCA` identifies 3 clusters. Similarly, on iNaturalist, `FedGWC` consistently surpasses FL baselines, reaching an average accuracy of 47.8% with $\beta = 0.5$ (automatically detecting a partition with 4 clusters). Results in Table 2 remark that, when dealing with realistic complex decentralized scenarios, standard aggregation methods are not able to mitigate the effects of heterogeneity across the federation, while, on the other hand, clustered FL provides a more efficient solution.

### 7.3. Clustering analysis of `FedGWC`

In this section, we investigate the underlying mechanisms behind `FedGWC` 's clustering in heterogeneous scenarios on Cifar100. Further experiments on Cifar10 are presented in Appendix H.

**`FedGWC` detects different client class distributions** We explore how the algorithm identifies and groups clients based on the non-IID nature of their data distributions, represented by the Dirichlet concentration parameter $\alpha$. For the Cifar100 dataset, we apply a similar splitting approach, obtaining the following partitions: (1) 90 clients with $\alpha = 0$ and 10 clients with $\alpha = 1000$; (2) 90 clients with $\alpha = 0.5$ and 10 clients with $\alpha = 1000$; and (3) 40 clients with $\alpha = 1000$, 30 clients with $\alpha = 0.05$, and 30 clients with $\alpha = 0$. We evaluate the outcome of this clustering experiment by means of WAS and WADB. Results in Table 3 show that `FedGWC` detects clusters groups clients according to the level of heterogeneity of the group.

**`FedGWC` detects different visual client domains.** Here, we focus on scenarios with nearly uniform class imbalance (high $\alpha$ values) but with different visual domains to investigate how `FedGWC` forms clusters in such settings. We incorporated various artificial domains (non-perturbed, noisy, and blurred images) Cifar100 dataset under homogeneous conditions ($\alpha = 100.00$). Our results demonstrate that `FedGWC` effectively clustered clients according to these distinct do-

| Dataset | FedGWC | CFL | IFCA | FedAvg | FedAvgM | FedProx | FairAvg |
|---|---|---|---|---|---|---|---|
| Google Landmarks | **57.4** ±0.3 | 40.5 ±0.2 | 49.4 ±0.3 | 40.5 ±0.2 | 36.4 ±1.3 | 40.2 ±0.6 | 39.0 ±0.3 |
| iNaturalist | **47.8** ±0.2 | 45.3 ±0.1 | 45.8 ±0.6 | 45.3 ±0.1 | 37.7 ±1.4 | 44.9 ±0.2 | 45.1 ±0.2 |

*Table 2.* Comparison of test accuracy on large scale FL datasets Google Landmarks and iNaturalist, between FedGWC and FL baselines – all clustered FL algorithms use FedAvg aggregation. FedGWC outperforms both clustered and standard FL methods detecting 5 and 4 clusters on Landmarks and iNaturalist, respectively.

*Table 3.* Clustering with three different splits on Cifar100 datasets. FedGWC has superior clustering quality across different splits (homogeneous $\alpha = 1000$, heterogeneous $\alpha = 0.05$, extremely heterogeneous $\alpha = 0$. )

| Dataset | (Hom, Het, X Het) | Clustering method | C | WAS↑ | WADB↓ |
|---|---|---|---|---|---|
| | | IFCA | 5 | -0.9 ± 0.0 | 1.8 ± 0.0 |
| | (10, 0, 90) | FeSem | 5 | -0.8 ± 0.2 | 2.6 ± 0.6 |
| | | FedGWC | 5 | **0.1 ± 0.1** | **0.2 ± 0.2** |
| Cifar100 | | IFCA | 5 | -0.0 ± 0.0 | **5.6 ± 1.5** |
| | (10, 90, 0) | FeSem | 5 | 0.2 ± 0.1 | 12.0 ± 2.0 |
| | | FedGWC | 5 | **0.4 ± 0.1** | 6.4 ± 2.0 |
| | | IFCA | 5 | -0.2 ± 0.0 | 1.0 ± 0.0 |
| | (40, 30, 30) | FeSem | 5 | -0.2 ± 0.0 | 33.2 ± 0.0 |
| | | FedGWC | 3 | **0.4 ± 0.2** | **0.9 ± 0.1** |

*Table 4.* Clustering performance of FedGWC is assessed on federations with clients from varied domains using clean, noisy, and blurred (Clean, Noise, Blur) images from Cifar100 datasets. It utilizes the Rand Index score (Rand, 1971), where a value close to 1 represents a perfect match between clustering and labels. Consistently FedGWC accurately distinguishes all visual domains. The ground truth number of clusters is respectively 2, 2, and 3.

| Dataset | (Clean, Noise, Blur) | Clustering method | C | Automatic Cluster Selection | Rand ↑ (max = 1.0) |
|---|---|---|---|---|---|
| | | IFCA | 1 | ✗ | 0.5 ± 0.0 |
| | (50, 50, 0) | FeSem | 2 | ✗ | 0.49 ± 0.2 |
| | | FedGWC | 2 | ✓ | **1.0 ± 0.0** |
| Cifar100 | | IFCA | 1 | ✗ | 0.5 ± 0.0 |
| | (50, 0, 50) | FeSem | 2 | ✗ | 0.51 ± 0.1 |
| | | FedGWC | 2 | ✓ | **1.0 ± 0.0** |
| | | IFCA | 1 | ✗ | 0.33 ± 0.0 |
| | (40, 30, 30) | FeSem | 3 | ✗ | 0.55 ± 0.0 |
| | | FedGWC | 4 | ✓ | **0.6 ± 0.0** |

mains. Table 4 presents the Rand-Index scores, which assess clustering quality based on known domain labels. The high Rand-Index scores, often approaching the upper bound of 1, indicate that FedGWC successfully separated clients into distinct clusters corresponding to their respective domains. This anaylsis suggests that FedGWC may be applicable for detecting malicious clients in FL, pinpointing a potential direction for future research.

## 8. Conclusions

We propose FedGWC, an efficient clustering algorithm for heterogeneous FL settings addressing the challenge of non-IID data and class imbalance. Unlike existing clustered FL methods, FedGWC groups clients by data distributions with flexibility and robustness, simply using the information encoded by the individual empirical loss. FedGWC successfully detects homogeneous clusters, as proved by our proposed novel Wasserstein Adjusted Score. FedGWC detects splits by removing out-of-distribution clients, thus simplifying the learning task within clusters without increasing communication overhead or computational cost. Empirical evaluations show that separately training classical FL algorithms on the homogeneous clusters detected by FedGWC consistently improves the performance. Additionally, FedGWC excels over other clustering techniques in grouping clients uniformly with respect to class imbalance and heterogeneity levels, which is crucial to mitigate the effect of non-IIDness FL. Finally, clustering on different class unbalanced and domain unbalanced scenarios, which are correctly detected by FedGWC (see Section 7.3), suggests that FedGWC can also be applied to anomaly client detection and to enhance robustness against malicious attacks in future research.

## Impact Statement

FedGWC is an efficient clustering-based approach that improves personalization while reducing communication and computation costs, enhancing the advance in the field of Federated Learning. Providing efficiency and explainability in clustering decisions, FedGWC enables more interpretable and scalable federated learning. Its efficiency makes it well-suited for IoT, decentralized AI, and sustainable AI applications, particularly in privacy-sensitive domains.

## Acknowledgements

We acknowledge the CINECA award under the ISCRA initiative for the availability of high-performance computing resources and support. A.L. worked under the auspices of Italian National Group of Mathematical Physics (GNFM) of INdAM, and was supported by the Project Piano Nazionale di Ripresa e Resilienza - Next Generation EU (PNRR-NGEU) from Italian Ministry of University and Research (MUR) under Grant DM 117/2023. This work was partially supported by the Innovation Grant gAIA within the activities of the National Research Center in High Performance Computing, Big Data and Quantum Computing (ICSC) CN00000013 - Spoke 1. Funded by the European

Union under Horizon Europe Programme - Grant Agreement 101123000 – Act.AI. Views and opinions expressed are however those of the author(s) only and do not necessarily reflect those of the European Union or European Research Council Executive Agency (ERCEA). Neither the European Union nor the granting authority can be held responsible for them.

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

## A. Theoretical Results for `FedGWC`

This section provides algorithms, in pseudo-code, to describe FedGWC (see Algorithms 2 and 1). Additionally, here we provide the proofs for the convergence results introduced in Section 5, specifically addressing the convergence (Theorems 5.1 and 5.2) and the formal derivation on the variance bound of the Gaussian weights (Proposition 5.3). In addition, we also present a sufficient condition, under which is guaranteed that the overall sampling rate of the training algorithm does not increase and remain unchanged during the training process (Theorem A.5).

**Theorem A.1.** *Let $\{\alpha_t\}_{t=1}^{\infty}$ be a sequence of positive real values, and $\{\Gamma_k^t\}_{t=1}^{\infty}$ the sequence of Gaussian weights. If $\{\alpha_t\}_{t=1}^{\infty} \in l^2(\mathbb{N})/l^1(\mathbb{N})$, then $\Gamma_k^t$ converges in $L^2$. Furthermore, for $t \to \infty$,*

$$\Gamma_k^t \longrightarrow \mu_k \quad a.s. \tag{8}$$

*Proof.* At each communication round, we compute the samples $r_i^{t,s}$ from $R_k^{t,s}$ via a Gaussian transformation of the observed loss in Eq. 1. Notice that, due to the linearity of the expectation operator, $\mathbb{E}[\Omega_k^t] = \mu_k$, that is the true, unknown, expected reward. The observed value for the random variable is given by $\omega_k^t = 1/S \sum_{s=1}^{S} r_k^{t,s}$, which is sampled from a distribution centered on $\mu_k$. Each client's weight is updated according to

$$\gamma_k^{t+1} = (1 - \alpha_t)\gamma_t + \alpha_t \omega_k^t . \tag{9}$$

Since such an estimator follows a Robbins-Monro algorithm, it is proved to converge in $L^2$. In addition, $\Gamma_k^t$ converges to the expectation $\mathbb{E}[\Omega_k^t] = \mu_k$ with probability 1, provided that $\alpha_t$ satisfies $\sum_{t \geq 1} |\alpha_t| = \infty$, and $\sum_{t \geq 1} |\alpha_t|^2 < \infty$ (Harold et al., 1997). $\square$

**Theorem A.2.** *Let $\alpha \in (0, 1)$ be a fixed constant, then in the limit $t \to \infty$, the expectation of the weights converges to the individual theoretical reward $\mu_k$, for each client $k = 1, \dots, K$, i.e.,*

$$\mathbb{E}[\Gamma_k^t] \longrightarrow \mu_k \quad t \to \infty . \tag{10}$$

*Proof.* Recall that $\gamma_k^{t+1} = (1 - \alpha)\gamma_k^t + \alpha\omega_k^t$, where $\omega_k^t$ are samples from $\Omega_k^t$. If we substitute backward the value of $\gamma_k^t$ we can write

$$\gamma_k^{t+1} = (1 - \alpha)^2 \gamma_k^{t-1} + \alpha\omega_k^t + \alpha(1 - \alpha)\omega_k^{t-1} . \tag{11}$$

By iterating up to the initialization term $\gamma_k^0$ we get the following formulation:

$$\gamma_k^{t+1} = (1 - \alpha)^{t+1}\gamma_k^0 + \sum_{\tau=0}^{t} \alpha(1 - \alpha)^{\tau}\omega_k^{t-\tau} . \tag{12}$$

Since $\omega_k^t$ are independent and identically distributed samples from $\Omega_k^t$, with expected value $\mu_k$, then the expectation of the weight at the $t$-th communication round would be

$$\mathbb{E}[\Gamma_k^t] = \mathbb{E}\left[(1 - \alpha)^t \gamma_k^0 + \sum_{\tau=0}^{t} \alpha(1 - \alpha)^{\tau}\Omega_k^{t-\tau-1}\right] , \tag{13}$$

that, due to the linearity of expectation, becomes

$$\mathbb{E}[\Gamma_k^t] = (1 - \alpha)^t \gamma_k^0 + \sum_{\tau=0}^{t} \alpha(1 - \alpha)^{\tau}\mu_k . \tag{14}$$

If we compute the limit

$$\lim_{t \to \infty} \mathbb{E}[\Gamma_k^t] = \lim_{t \to \infty} (1 - \alpha)^t \gamma_k^0 + \sum_{\tau=0}^{\infty} \alpha(1 - \alpha)^{\tau}\mu_k , \tag{15}$$

and since $\alpha \in (0, 1)$, the first term tends to zero, and also the geometric series converges. Therefore, the expectation of the weights converges to $\mu_k$, namely

$$\lim_{t \to \infty} \mathbb{E}[\Gamma_k^t] = \mu_k . \tag{16}$$

$\square$

**Proposition A.3.** *The variance of the weights $\Gamma_k^t$ is smaller than the variance $\sigma_k^2$ of the theoretical rewards $R_k^{t,s}$.*

*Proof.* From Eq.12, we can show that $\mathbb{V}ar(\Gamma_k^t)$ converges to a value that depends on $\alpha$ and the number of local training iterations $S$. Indeed

$$
\mathbb{V}ar(\Gamma_k^t) = \mathbb{V}ar\left((1-\alpha)^t\gamma_k^0 + \sum_{\tau=0}^{t}\alpha(1-\alpha)^\tau\Omega_k^{t-\tau-1}\right)
$$
$$
= \sum_{\tau=0}^{t}\alpha^2(1-\alpha)^{2\tau}\mathbb{V}ar(\Omega_k^t) = \frac{1}{S}\sum_{\tau=0}^{t}\alpha^2(1-\alpha)^{2\tau}\sigma_k^2
\tag{17}
$$

since $\Omega_k^t = 1/S\sum_{s=1}^{S}R_k^{t,s}$.

If we compute the limit, that exists finite due to the hypothesis $\alpha \in (0,1)$, we get

$$
\lim_{t\to\infty}\mathbb{V}ar(\Gamma_k^t) = \frac{\alpha^2\sigma_k^2}{S}\sum_{\tau=0}^{\infty}(1-\alpha)^{2\tau} = \frac{\alpha}{2-\alpha}\frac{\sigma_k^2}{S} < \frac{\sigma_k^2}{S} < \sigma_k^2 .
\tag{18}
$$

$\square$

We further demonstrate that the interaction matrix $P^t$ identified by `FedGWC` is entry-wise bounded from above, as established in the following proposition.

**Proposition A.4.** *The entries of the interaction matrix $P^t$ are bounded from above, namely for any $t \geq 0$ there exists a positive finite constant $C_t > 0$ such that*

$$
P_{kj}^t \leq C_t .
\tag{19}
$$

*And furthermore*

$$
\lim_{t\to\infty}C_t = 1 .
\tag{20}
$$

*Proof.* Without loss of generality we assume that every client of the federation is sampled, and we assume that $\alpha_t = \alpha \in (0,1)$ for any $t \geq 0$. We recall, from Eq.3, that for any couple of clients $k, j \in \mathcal{P}_t$ the entries of the interaction matrix are updated according to

$$
P_{kj}^{t+1} = (1-\alpha)P_{kj}^t + \alpha\omega_k^t .
\tag{21}
$$

If we iterate backward until $P_{kj}^0$, we obtain the following update

$$
P_{kj}^{t+1} = (1-\alpha)^{t+1}P_{kj}^0 + \sum_{\tau=0}^{t}\alpha(1-\alpha)^\tau\omega_k^{t-\tau} .
\tag{22}
$$

We know that, by constructions, the Gaussian rewards $\omega_k^t < 1$ at any time $t$, therefore the following inequality holds

$$
P_{kj}^t = (1-\alpha)^tP_{kj}^0 + \sum_{\tau=0}^{t}\alpha(1-\alpha)^\tau\omega_k^{t-\tau-1} \leq (1-\alpha)^tP_{kj}^0 + \sum_{\tau=0}^{t}\alpha(1-\alpha)^\tau .
\tag{23}
$$

At any round $t$ we can define the constant $C_t$, as

$$
C_t := (1-\alpha)^tP_{kj}^0 + \alpha\sum_{\tau=0}^{t}(1-\alpha)^\tau = (1-\alpha)^tP_{kj}^0 + 1 - (1-\alpha)^{t+1} < \infty .
\tag{24}
$$

Moreover, since $\alpha \in (0,1)$, by taking the limit we prove that

$$
\lim_{t\to\infty}C_t = \lim_{t\to\infty}(1-\alpha)^tP_{kj}^0 + 1 - (1-\alpha)^{t+1} = 1 .
\tag{25}
$$

$\square$

---

**Algorithm 2** `FedGW_Cluster`

---

1: **Input:** $P, n_{max}, \mathcal{K}(\cdot, \cdot)$
2: **Output:** cluster labels $y_{n_{cl}}$, and number of clusters $n_{cl}$
3: Extract UPVs $v_k^j, v_j^k$ from $P$ for any $k, j$
4: $W_{kj} \leftarrow \mathcal{K}(v_k^j, v_j^k)$ for any $k, j$
5: **for** $n = 2, \ldots, n_{max}$ **do**
6:   $y_n \leftarrow$ `Spectral_Clustering`$(W, n)$
7:   $DB_n \leftarrow$ `Davies_Bouldin`$(W, y_n)$
8:   **if** $\min_n DB_n > 1$ **then**
9:     $n_{cl} \leftarrow 1$
10:   **else**
11:     $n_{cl} \leftarrow \arg\min_n DB_n$
12:   **end if**
13: **end for**

---

**Theorem A.5.** *(Sufficient Condition for Sample Rate Conservation) Consider $K_{min}$ as the minimum number of clients permitted per cluster, i.e. the cardinality $|\mathcal{C}_n| \geq K_{min}$ for any given cluster $n = 1, \ldots, n_{cl}$, and $\rho \in (0, 1]$ to represent the initial sample rate. There exists a critical threshold $n^* > 0$ such that, if $K_{min} \geq n^*$ is met, the total sample size does not increase.*

*Proof.* Let us denote by $\rho_n$ the participation rate relative to the $n$-th cluster, *i.e.*

$$\rho_n = \max\left\{\rho, \frac{3}{|\mathcal{C}_n|}\right\} \tag{26}$$

because, in order to maintain privacy of the clients' information we need to sample at least three clients, therefore $\rho^n$ is at least 3 over the number of clients belonging to the cluster. The total participation rate at the end of the clustering process is given by

$$\rho^{\text{global}} = \sum_{n=1}^{n_{cl}} \frac{K_n}{K} \tag{27}$$

where $K_n$ denotes the number of clients sampled within the $n$-th cluster. If we focus on the term $K_n$, recalling Equation 26, we have that

$$K_n = \rho_n |\mathcal{C}_n| = \max\left\{\rho, \frac{3}{|\mathcal{C}_n|}\right\} \times |\mathcal{C}_n| = \max\{\rho|\mathcal{C}_n|, 3\} . \tag{28}$$

If we write Equation 28, by the means of the positive part function, denoted by $(x)^+ = \max\{0, x\}$, we obtain that

$$K_n = 3 + \max\{0, \rho|\mathcal{C}_n| - 3\} = 3 + (\rho|\mathcal{C}_n| - 3)^+ . \tag{29}$$

Observe that we are looking for a threshold value for which $\rho^{\text{global}} = \rho$, *i.e.* the participation rate remains the same during the whole training process.

Let us observe that $K_n = \rho|\mathcal{C}_n| \iff \rho|\mathcal{C}_n| \geq 3 \iff |\mathcal{C}_n| \geq n^* = 3/\rho$. In fact, if we assume that $K_{min} \geq n^*$, then the following chain of equalities holds

$$\rho^{\text{global}} = \sum_{n=1}^{n_{cl}} \frac{K_n}{K} = \frac{1}{K} \sum_{n=1}^{n_{cl}} \rho|\mathcal{C}_n| = \frac{\rho}{K} \sum_{n=1}^{n_{cl}} |\mathcal{C}_n| = \frac{\rho K}{K} = \rho$$

thus proving that $K_{min} \geq n^*$ is a sufficient condition for not increasing the sampling rate during the training process. $\square$

## B. Theoretical Derivation of the Wasserstein Adjusted Score

To address the lack of clustering evaluation metrics suited for FL with distributional heterogeneity and class imbalance, we introduced a theoretically grounded adjustment to standard metrics, derived from the Wasserstein distance, Kantorovich–Rubinstein metric (Kantorovich, 1942). This metric, integrated with popular scores like Silhouette and Davies-Bouldin, enables a modular framework for a posteriori evaluation, effectively comparing clustering outcomes across federated algorithms. In this paragraph, we show how the proposed clustering metric that accounts for class imbalance can be derived from a probabilistic interpretation of clustering.

**Definition B.1.** Let $(M, d)$ be a metric space, and $p \in [1, \infty]$. The Wasserstein distance between two probability measures $\mathbb{P}$ and $\mathbb{Q}$ over $M$ is defined as

$$W_p(\mathbb{P}, \mathbb{Q}) = \inf_{\gamma \in \Gamma(\mathbb{P}, \mathbb{Q})} \mathbb{E}_{(x,y) \sim \gamma}[d(x, y)^p]^{1/p} \tag{30}$$

where $\Gamma(\mathbb{P}, \mathbb{Q})$ is the set of all the possible couplings of $\mathbb{P}$ and $\mathbb{Q}$ (see Def. B.2).

Furthermore, we need to introduce the notion of coupling of two probability measures.

**Definition B.2.** Let $(M, d)$ be a metric space, and $\mathbb{P}, \mathbb{Q}$ two probability measures over $M$. A coupling $\gamma$ of $\mathbb{P}$ and $\mathbb{Q}$ is a joint probability measure on $M \times M$ such that, for any measurable subset $A \subset M$,

$$\begin{aligned}
\int_A \left( \int_M \gamma(dx, dy) \mathbb{Q}(dy) \right) \mathbb{P}(dx) &= \mathbb{P}(A), \\
\int_A \left( \int_M \gamma(dx, dy) \mathbb{P}(dx) \right) \mathbb{Q}(dy) &= \mathbb{Q}(A).
\end{aligned} \tag{31}$$

Let us recall that the empirical measure over $M$ of a sample of observations $\{x_1, \cdots, x_N\}$ is defined such that for any measurable set $A \subset M$

$$\mathbb{P}(A) = \frac{1}{N} \sum_{i=1}^{C} \delta_{x_i}(A) \tag{32}$$

where $\delta_{x_i}$ is the Dirac's measure concentrated on the data point $x_i$.

In particular, we aim to measure the goodness of a cluster by taking into account the distance between the empirical frequencies between two clients' class distributions and use that to properly adjust the clustering metric. For the sake of simplicity, we assume that the distance $d$ over $M$ is the $L^2$-norm. We obtain the following theoretical result to justify the rationale behind our proposed metric.

**Theorem B.3.** *Let $s$ be an arbitrary clustering score. Then, the class-imbalance adjusted score $\tilde{s}$ is exactly the metric $s$ computed with the Wasserstein distance between the empirical measures over each client's class distribution.*

*Proof.* Let us consider two clients; each one has its own sample of observations $\{x_1, \ldots, x_C\}$ and $\{y_1, \ldots, y_C\}$ where the $i$-th position corresponds to the frequency of training points of class $i$ for each client. We aim to compute the $p$-Wasserstein distance between the empirical measures $\mathbb{P}$ and $\mathbb{Q}$ of the two clients, in particular for any $dx, dy > 0$

$$\begin{aligned}
\mathbb{P}(dx) &= \frac{1}{N} \sum_{i=1}^{N} \delta_{x_i}(dx), \\
\mathbb{Q}(dy) &= \frac{1}{N} \sum_{i=1}^{N} \delta_{y_i}(dy) \ .
\end{aligned} \tag{33}$$

In order to compute $W_p^p(\mathbb{P}, \mathbb{Q})$ we need to carefully investigate the set of all possible coupling measures $\Gamma(\mathbb{P}, \mathbb{Q})$. However, since either $\mathbb{P}$ and $\mathbb{Q}$ are concentrated over countable sets, it is possible to see that the only possible couplings satisfying Eq. 31 are the Dirac's measures over all the possible permutations of $x_i$ and $y_i$. In particular, by fixing the ordering of $x_i$, according to the rank statistic $x_{(i)}$, the coupling set can be written as

$$\Gamma(\mathbb{P}, \mathbb{Q}) = \left\{ \frac{1}{C} \delta_{(x_{(i)}, y_{\pi(i)})} : \pi \in \mathcal{S} \right\} \tag{34}$$

where $\mathcal{S}$ is the set of all possible permutations of $C$ elements. Therefore we could write Eq. 30 as follows

$$W_p^p = \min_{\pi \in \mathcal{S}} \int_{M \times M} |x - y|^p \frac{1}{N} \sum_{i=1}^{C} \delta_{(x_{(i)}, y_{\pi(i)})}(dx, dy) \tag{35}$$

since $\mathcal{S}$ is finite, the infimum is a minimum. By exploiting the definition of Dirac's distribution and the linearity of the Lebesgue integral, for any $\pi \in \mathcal{S}$, we get

$$\int_{M \times M} |x - y|^p \frac{1}{C} \sum_{i=1}^{C} \delta_{(x_{(i)}, y_{\pi(i)})}(dx, dy) = \frac{1}{C} \sum_{i=1}^{C} \int_{M \times M} |x - y|^p \delta_{(x_{(i)}, y_{\pi(i)})}(dx, dy)$$
$$= \frac{1}{C} \sum_{i=1}^{C} |x_{(i)} - y_{\pi(i)}|^p . \tag{36}$$

Therefore, finding the Wasserstein distance between $\mathbb{P}$ and $\mathbb{Q}$ boils down to a combinatorial optimization problem, that is, finding the permutation $\pi \in \mathcal{S}$ that solves

$$W_p^p(\mathbb{P}, \mathbb{Q}) = \min_{\pi \in \mathcal{S}} \frac{1}{C} \sum_{i=1}^{C} |x_{(i)} - y_{\pi(i)}|^p . \tag{37}$$

The minimum is achieved when $\pi = \pi^*$ that is the permutation providing the ranking statistic, i.e. $\pi^*(y_i) = y_{(i)}$, since the smallest value of the sum is given for the smallest fluctuations. Thus we conclude that the $p$-Wasserstein distance between $\mathbb{P}$ and $\mathbb{Q}$ is given by

$$W_p(\mathbb{P}, \mathbb{Q}) = \left( \frac{1}{C} \sum_{i=1}^{C} |x_{(i)} - y_{(i)}|^p \right)^{1/p} \tag{38}$$

that is the pairwise distance computed between the class frequency vectors, sorted in order of magnitude, for each client, introduced in Section 6, where we chose $p = 2$. $\qquad\square$

## C. Privacy of `FedGWC`

In the framework of `FedGWC`, clients are required to send only the empirical loss vectors $l_k^{t,s}$ to the server (Cho et al., 2022). While concerns might arise regarding the potential leakage of sensitive information from sharing this data, it is important to clarify that the server only needs to access aggregated statistics, working on aggregated data. This ensures that client-specific information remains private. Privacy can be effectively preserved by implementing the Secure Aggregation protocol (Bonawitz et al., 2016), which guarantees that only the aggregated results are shared, preventing the exposure of any raw client data.

## D. Communication and Computational Overhead of `FedGWC`

`FedGWC` minimizes communication and computational overhead, aligning with the requirements of scalable FL systems (McMahan et al., 2016). On the client side, the computational cost remains unchanged compared to the chosen FL aggregation, e.g. FedA, as clients are only required to communicate their local models and a vector of empirical losses after each round. The size of this loss vector, denoted by $S$, corresponds to the number of local iterations (*i.e.* the product of local epochs and the number of batches) and is negligible w.r.t. the size of the model parameter space, $|\Theta|$. In our experimental setup, $S = 8$, ensuring that the additional communication overhead from transmitting loss values is negligible in comparison to the transmission of model weights.

All clustering computations, including those based on interaction matrices and Gaussian weighting, are performed exclusively on the server. This design ensures that client devices are not burdened with additional computational complexity or memory demands. The interaction matrix $P$ used in `FedGWC` is updated incrementally and involves sparse matrix operations, which significantly reduce both memory usage and computational costs.

These characteristics make `FedGWC` particularly well-suited for cross-device scenarios involving large federations and numerous communication rounds. Moreover, by operating on scalar loss values rather than high-dimensional model parameters, the clustering process in `FedGWC` achieves computational efficiency while maintaining effective grouping

of clients. The server-side processing ensures that the method remains scalable, even as the number of clients and communication rounds increases. Consequently, FedGWC meets the fundamental objectives of FL by minimizing costs while preserving privacy and maintaining high performance.

**Computational Costs Analysis** In this section, we present the analysis of computational costs for FedGWC. We would like to emphasize that all additional computational costs are incurred on the server-side, which in typical FL frameworks possesses abundant resources. Importantly, our approach does not increase the computational burden on the client side beyond the standard cost of the chosen FL aggregation method (e.g., backpropagation for FedAvg).

The most computationally intensive operation in our clustering procedure is spectral clustering. For an interaction matrix involving $K$ clients, where $m$ is the number of eigenvectors considered (i.e., the number of clusters sought) and $H$ is the number of k-means iterations required within spectral clustering, a single execution of spectral clustering has a complexity of $\mathcal{O}(Km^2 + KmH)$ (Von Luxburg, 2007). To determine the best partition of the client federation, we evaluate clustering outcomes for a varying number of clusters $m$, from $m = 2, \ldots, n_{\max}$, where $n_{\max}$ is the maximum number of clusters admitted. This means spectral clustering is performed $n_{\max} - 1$ times. The total complexity for this phase is the sum of costs for each $m$:

$$\sum_{m=2}^{n_{\max}} \left( \mathcal{O}(Km^2 + KmH) \right) = \mathcal{O}(Kn_{\max}^3 + KHn_{\max}^2).$$

This entire clustering phase is performed only a limited number of times during training (typically three to four times on average in our experiments); this small constant factor is absorbed by the $\mathcal{O}$ notation. In our experimental settings, $H$ was 300, and $n_{\max}$ (the maximum number of clusters explored) ranged between 5 and 10. The parameters $n_{\max}$ and $H$ are fixed for each invocation of the clustering phase.

Beyond clustering, other computational steps on the server at each communication round $t$ include:

- **Computing client weights** $\omega_k^t$ has a cost of $\mathcal{O}(|\mathcal{P}_t|S)$, where $|\mathcal{P}_t|$ is the number of participating clients at round $t$, and $S$ is the number of local iterations performed by clients. This cost reflects server-side work proportional to the number of participants and their local effort.

- **Other server-side computations**, for instance, operations like computing global sample means and variances from aggregated client information might contribute an additional $\mathcal{O}(S)$ per round.

Thus, the total cost for these operations over $T$ communication rounds is $\mathcal{O}(T|\mathcal{P}_t|S + TS)$. In our setting, $|\mathcal{P}_t|$ was typically 10 and $S = 8$.

Considering that the clustering (costing $\mathcal{O}(Kn_{\max}^3 + KHn_{\max}^2)$) occurs only a few times, the overall additional computational overhead introduced by our method can be expressed as:

$$\mathcal{O}(Kn_{\max}^3 + KHn_{\max}^2) + \mathcal{O}(T|\mathcal{P}_t|S + TS). \tag{39}$$

By applying properties of Landau notation and considering parameters $n_{\max}$, $H$, $S$, and the client sampling rate $\rho = |\mathcal{P}_t|/K$ as constants or small fixed values (which do not scale with $K$ or $T$, a common consideration in large-scale FL systems (Bonawitz et al., 2019)), the overall computational cost can be simplified. Substituting $|\mathcal{P}_t| = \rho K$ into the expression:

$$\mathcal{O}(Kn_{\max}^3 + KHn_{\max}^2 + T\rho KS + TS).$$

This simplifies with respect to the total number of clients $K$ and the total number of communication rounds $T$:

$$\mathcal{O}(K) + \mathcal{O}(K) + \mathcal{O}(TK) + \mathcal{O}(T) = \mathcal{O}(TK + T + K).$$

This analysis demonstrates that FedGWC not only places all computationally intensive operations on the server but also introduces a very limited overall computational cost, primarily scaling with $TK$.

## E. Metrics Used for Evaluation
### E.1. Silhouette Score
Silhouette Score is a clustering metric that measures the consistency of points within clusters by comparing intra-cluster and nearest-cluster distances (Rousseeuw, 1987). Let us consider a metric space $(M, d)$. For a set of points $\{x_1, \ldots, x_N\} \subset M$

and clustering labels $\mathcal{C}_1, \ldots, \mathcal{C}_{n_{cl}}$. The Silhouette score of a data point $x_i$ belonging to a cluster $\mathcal{C}_i$ is defined as

$$s_i = \frac{b_i - a_i}{\max\{a_i, b_i\}} \tag{40}$$

where the values $b_i$ and $a_i$ represent the average intra-cluster distance and the minimal average outer-cluster distance, *i.e.*

$$a_i = \frac{1}{|\mathcal{C}_i| - 1} \sum_{x_j \in \mathcal{C}_i \setminus \{x_i\}} d(x_i, x_j)$$

$$b_i = \min_{j \neq i} \frac{1}{|\mathcal{C}_j|} \sum_{x_j \in \mathcal{C}_j} d(x_i, x_j) \tag{41}$$

The value of the Silhouette score ranges between $-1$ and $+1$, *i.e.* $s_i \in [-1, 1]$. In particular, a Silhouette score close to 1 indicates well-clustered data points, 0 denotes points near cluster boundaries, and -1 suggests misclassified points. In order to evaluate the overall performance of the clustering, a common choice, that is the one adopted in this paper, is to average the score value for each data point.

### E.2. Davies-Bouldin Score
The Davies-Bouldin Score is a clustering metric that evaluates the quality of clustering by measuring the ratio of intra-cluster dispersion to inter-cluster separation (Davies & Bouldin, 1979). Let us consider a metric space $(M, d)$, a set of points $\{x_1, \ldots, x_N\} \subset M$, and clustering labels $\mathcal{C}_1, \ldots, \mathcal{C}_{n_{cl}}$. The Davies-Bouldin score is defined as the average similarity measure $R_{ij}$ between each cluster $\mathcal{C}_i$ and its most similar cluster $\mathcal{C}_j$:

$$DB = \frac{1}{n_{cl}} \sum_{i=1}^{n_{cl}} \max_{j \neq i} R_{ij} \tag{42}$$

where $R_{ij}$ is given by the ratio of intra-cluster distance $S_i$ to inter-cluster distance $D_{ij}$, *i.e.*

$$R_{ij} = \frac{S_i + S_j}{D_{ij}} \tag{43}$$

with intra-cluster distance $S_i$ defined as

$$S_i = \frac{1}{|\mathcal{C}_i|} \sum_{x_k \in \mathcal{C}_i} d(x_k, c_i) \tag{44}$$

where $c_i$ denotes the centroid of cluster $\mathcal{C}_i$, and $D_{ij} = d(c_i, c_j)$ is the distance between centroids of clusters $\mathcal{C}_i$ and $\mathcal{C}_j$. A lower Davies-Bouldin Index indicates better clustering, as it reflects well-separated and compact clusters. Conversely, a higher DBI suggests that clusters are less distinct and more dispersed.

### E.2.1. RAND INDEX
Rand Index is a clustering score that measures the outcome of a clustering algorithm with respect to a ground truth clustering label (Rand, 1971). Let us denote by $a$ the number of pairs that have been grouped in the same clusters, while by $b$ the number of pairs that have been grouped in different clusters, then the Rand-Index is defined as

$$RI = \frac{a + b}{\binom{N}{2}} \tag{45}$$

where N denotes the number of data points. In our experiments we opted for the Rand Index score to evaluate how the algorithm was able to separate clients in groups of the same level of heterogeneity (which was known a priori and used as ground truth). A Rand Index ranges in $[0, 1]$, and a value of 1 signifies a perfect agreement between the identified clusters and the ground truth.

## F. Datasets and implementation details
To simulate a realistic FL environment with heterogeneous data distribution, we conduct experiments on Cifar100 (Krizhevsky et al., 2009). As a comparison, we also run experiments on the simpler Cifar10 dataset (Krizhevsky et al., 2009).

Cifar10 and Cifar100 are distributed among $K$ clients using a Dirichlet distribution (by default, we use $\alpha = 0.05$ for Cifar10 and $\alpha = 0.5$ for Cifar100) to create highly imbalanced and heterogeneous settings. By default, we use $K = 100$ clients with 500 training and 100 test images. The classification model is a CNN with two convolutional blocks and three dense layers. Additionally, we perform experiments on the Femnist dataset (LeCun, 1998), partitioned among 400 clients using a Dirichlet distribution with $\alpha = 0.01$. The choice of different values for $\alpha$ is made to ensure comparable levels of heterogeneity through the adjustment of the Dirichlet parameter in relation to the number of classes. In particular, by employing $\alpha = 0.5$ for Cifar100 and $\alpha = 0.01$ for FeMNIST, we preserve a consistent ratio of $\alpha/C$ across the datasets. This approach aids in regulating the level of class imbalance, thereby ensuring comparability in heterogeneity across different datasets. In these experiments, we employ LeNet5 as the classification model (LeCun et al., 1998). Local training on each client uses SGD with a learning rate of 0.01, weight decay of $4 \cdot 10^{-4}$, and batch size 64. The number of local epochs is 1, resulting in 7 batch iterations for Cifar10 and Cifar100 and 8 batch iterations for Femnist. The number of communication rounds is set to 3,000 for Femnist, 10,000 for Cifar10 and 20,000 for Cifar100, with a 10% client participation rate per cluster. For FedGWC we tuned the hyper-parameter $\beta \in \{0.1, 0.5, 1, 2, 4\}$, *i.e.* the spread of the RBF kernel, and we set the tolerance $\epsilon$ to $10^{-5}$, constant value $\alpha_t = \alpha$ equal to the participation rate, *i.e.* 10%. FeSEM's and IFCA's number of clusters was tuned between 2,3,4, and 5. Each client has its own local training and test sets. We evaluate classification performance using balanced accuracy, computed per client as the average class-wise recall. The overall federated balanced accuracy is then obtained by averaging client-wise balanced accuracy, optionally weighted by test set sizes, to account for heterogeneous data distributions.

Large Scale experiments are conducted on Google Landmarks (Weyand et al., 2020) with $K = 823$ clients and (Van Horn et al., 2018) with $K = 2714$ clients, as partitioned in (Hsu et al., 2020). For Landmarks and iNaturalist, we always refer to the Landmark-Users-160K and iNaturalist-Users-120K partition, respectively. The classification model is MobileNetV2 architecture (Sandler et al., 2018) with pre-trained weights on ImageNet-1K dataset (Deng et al., 2009) optimized with SGD having learning rate of 0.1. To mimic real world low client availability we employed 10 sampled clients per communication round, with a total training of 1000 and 2000 communication rounds, with 7 and 5 batch iterations respectively. For FedGWC we tuned the hyper-parameter $\beta \in \{0.1, 0.5, 1, 2, 4\}$, *i.e.* the spread of the RBF kernel, and we set the tolerance $\epsilon$ to $10^{-2}$ for iNaturalist and to $10^{-4}$ for Landmark, constant value $\alpha_t = \alpha = .1$. IFCA's number of clusters was tuned between 2,3,4, and 5. Each client has its own local training and test sets. Performance in large scale scenarios are evaluated by averaging the accuracy achieved on the local test sets across the federation.

## G. Sensitive Analysis beta value RBF kernel

This section provides a sensitivity analysis for the $\beta$ hyper-parameter of the RBF kernel adopted for FedGWC. The results of this tuning are shown in Table 5.

*Table 5.* A sensitivity analysis on the RBF kernel hyper-parameter $\beta$ is conducted. We present the balanced accuracy for FedGWC on the Cifar10, Cifar100, and Femnist datasets for $\beta \in \{0.1, 0.5, 1.0, 2.0, 4.0\}$. It is noteworthy that FedGWC demonstrates robustness to variations in this hyperparameter.

| $\beta$ | Cifar100 | Femnist | Google Landmarks | iNaturalist |
|---|---|---|---|---|
| 0.1 | 49.9 | 76.0 | 55.0 | 47.5 |
| 0.5 | **53.4** | 76.0 | **57.4** | **47.8** |
| 1.0 | 49.5 | 76.0 | 56.0 | 47.5 |
| 2.0 | 50.9 | 75.6 | 57.0 | 47.2 |
| 4.0 | 52.6 | **76.1** | 55.8 | 47.1 |

## H. Additional Experiments: Visual Domain Detection in Cifar10

In this section we present the result for the domain ablation discussed in Section 7.3 conducted on Cifar10 (Krizhevsky et al., 2009). We explore how the algorithm identifies and groups clients based on the non-IID nature of their data distributions, represented by the Dirichlet concentration parameter $\alpha$. We apply a similar splitting approach, obtaining the following partitions: (1) 90 clients with $\alpha = 0$ and 10 clients with $\alpha = 100$; (2) 90 clients with $\alpha = 0.5$ and 10 clients with $\alpha = 100$; and (3) 40 clients with $\alpha = 100$, 30 clients with $\alpha = 0.5$, and 30 clients with $\alpha = 0$. We evaluate the outcome of this clustering experiment by means of WAS and WADB. Results in Table 3 show that FedGWC detects clusters groups clients

*Table 6.* Clustering with three different splits on Cifar10. `FedGWC` has superior clustering quality across different splits (homogeneous *Hom*, heterogeneous *Het*, extremly heterogeneous *X Het*)

| Dataset | (Hom, Het, X Het) | Clustering method | C | WAS | WADB |
|---------|-------------------|-------------------|---|-----|------|
| Cifar10 | (10, 0, 90) | IFCA | 1 | / | / |
| | | FeSem | 3 | $-0.0 \pm 0.1$ | $12.0 \pm 2.0$ |
| | | FedGWC | 3 | $\mathbf{0.1} \pm \mathbf{0.0}$ | $\mathbf{0.2} \pm \mathbf{0.0}$ |
| | (10, 90, 0) | IFCA | 1 | / | / |
| | | FeSem | 3 | $-0.0 \pm 0.0$ | $12.0 \pm 2.0$ |
| | | FedGWC | 3 | $\mathbf{0.2} \pm \mathbf{0.0}$ | $\mathbf{0.6} \pm \mathbf{0.0}$ |
| | (40, 30, 30) | IFCA | 2 | $-0.2 \pm 0.0$ | $\mathbf{1.0} \pm \mathbf{0.0}$ |
| | | FeSem | 3 | $0.1 \pm 0.1$ | $20.6 \pm 7.1$ |
| | | FedGWC | 3 | $\mathbf{0.6} \pm \mathbf{0.1}$ | $\mathbf{1.0} \pm \mathbf{0.4}$ |

*Table 7.* Clustering performance of `FedGWC` is assessed on federations with clients from varied domains using clean, noisy, and blurred (Clean, Noise, Blur) images from Cifar10 dataset. It utilizes the Rand Index score (Rand, 1971), where a value close to 1 represents a perfect match between clustering and labels. Consistently `FedGWC` accurately distinguishes all visual domains.

| Dataset | (Clean, Noise, Blur) | Clustering method | C | Rand |
|---------|----------------------|-------------------|---|------|
| Cifar10 | (50, 50, 0) | IFCA | 1 | $0.5 \pm 0.0$ |
| | | FeSem | 2 | $0.49 \pm 0.2$ |
| | | FedGWC | 2 | $\mathbf{1.0} \pm \mathbf{0.0}$ |
| | (50, 0, 50) | IFCA | 1 | $0.5 \pm 0.0$ |
| | | FeSem | 2 | $0.5 \pm 0.1$ |
| | | FedGWC | 2 | $\mathbf{1.0} \pm \mathbf{0.0}$ |
| | (40, 30, 30) | IFCA | 1 | $0.33 \pm 0.0$ |
| | | FeSem | 3 | $0.34 \pm 0.1$ |
| | | FedGWC | 4 | $\mathbf{0.9} \pm \mathbf{0.0}$ |

according to the level of heterogeneity of the group.

# I. Evaluation of IFCA and FeSEM algorithms with different number of clusters

This section shows the tuning of the number of clusters for the `IFCA` and `FeSEM` algorithms, which cannot automatically detect this value. The results of this tuning are shown in Table 8.

# J. Further Experiments

In Table 9 we show that `FedGWC` is orthogonal to FL aggregation, which means that any algorithm - including personalized FL algorithms - can be easily embedded in our clustering setting, improving the final performance. We show this in Table 9, where we also include personalization algorithms – `pFedMe` (T Dinh et al., 2020) and `Per-FedAvg` (Fallah et al., 2020).

Note that `FedGWC` does not directly compare to personalization methods, as it would result in unfair comparison, since personalization methods require more local resources and overhead than clustering methods. Personalization methods aim to create models uniquely tailored to each client, a distinct objective from clustering, which instead aims to identify and exploit shared structures within client clusters. Given these fundamental distinctions in their objectives – highly individualized models versus robust cluster level representations – and their differing resource demands, a direct benchmark of their standalone effectiveness would be misleading. Therefore, exploring their potential integration is a more insightful line of investigation. We rather show that applying `FedGWC` on top of personalization algorithms allows to increase model performance. Figure 5 illustrates the clustering results corresponding to varying degrees of heterogeneity, as described in Section 7.3. As per `FedGWC`, the detection of clusters based on different levels of heterogeneity in the Cifar10 dataset is achieved. Specifically, an examination of the interaction matrix reveals a clear distinction between the two groups. In Figure 6, we show that in class-balanced scenarios with small heterogeneity, like Cifar10 with $\alpha = 100$, `FedGWC` successfully detects one single cluster. Indeed, in homogeneous scenarios such as this one, the model benefits from accessing more data from all the clients.

Figure 4 shows how the MSE converges to a small value as the rounds increase for a Cifar10 experiment.

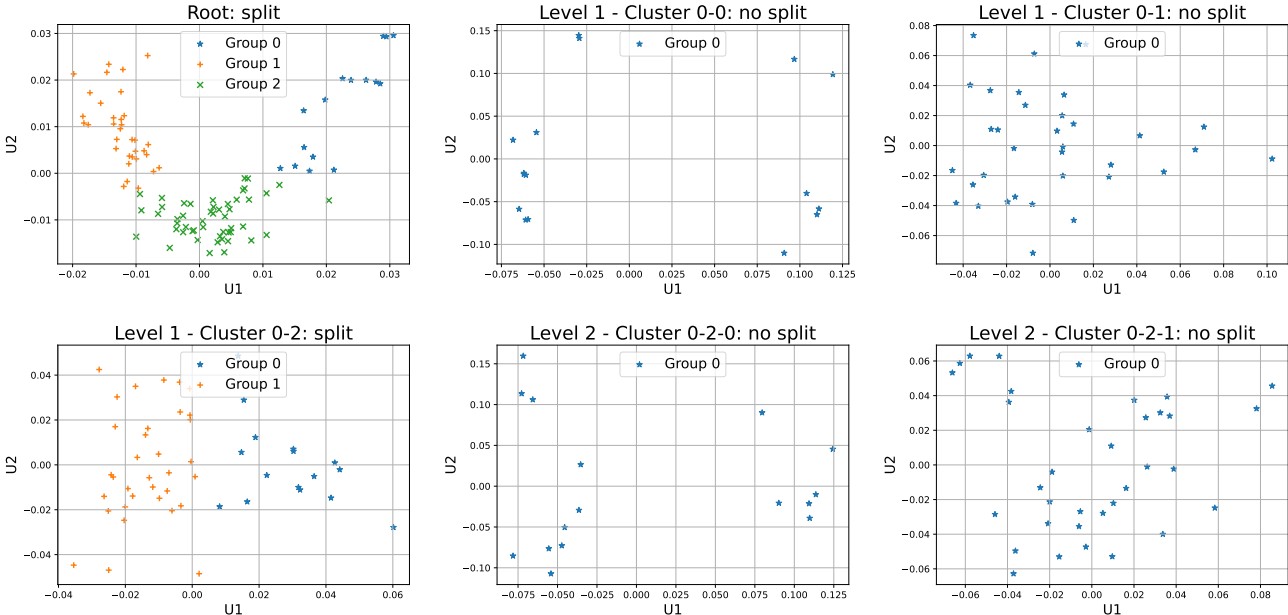

*Figure 3.* Cluster evolution with respect to the recursive splits in `FedGWC` on Cifar100, projected on the spectral embedded bi-dimensional space. From left to right, top to bottom, we can see that `FedGWC` splits the client into cluster, until a certain level of intra-cluster homogeneity is reached

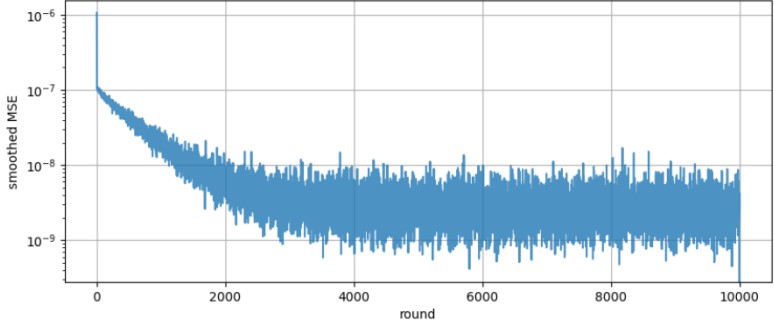

*Figure 4.* Interaction matrix convergence: on the $y$-axis MSE in logarithmic scale w.r.t. communication rounds in the $x$-axis on Cifar10, with Dirichlet parameter $\alpha = 0.05$.

*Table 8.* Performance of for baseline algorithms for clustering in FL `FeSEM`, and `IFCA`, w.r.t. the number of clusters

|  | Clustering method | C | Acc |
|---|---|---|---|
| Cifar100 | IFCA | 2 | $46.7 \pm 0.0$ |
|  |  | 3 | $44.0 \pm 1.6$ |
|  |  | 4 | $45.1 \pm 2.6$ |
|  |  | 5 | $47.5 \pm 3.5$ |
|  | FeSem | 2 | $43.3 \pm 1.3$ |
|  |  | 3 | $48.0 \pm 1.9$ |
|  |  | 4 | $50.9 \pm 1.8$ |
|  |  | 5 | $53.4 \pm 1.8$ |
| Femnist | IFCA | 2 | $76.1 \pm 0.1$ |
|  |  | 3 | $75.9 \pm 1.9$ |
|  |  | 4 | $76.6 \pm 0.1$ |
|  |  | 5 | $76.7 \pm 0.6$ |
|  | FeSem | 2 | $75.6 \pm 0.2$ |
|  |  | 3 | $75.5 \pm 0.5$ |
|  |  | 4 | $75.0 \pm 0.1$ |
|  |  | 5 | $74.9 \pm 0.1$ |

*Table 9.* `FedGWC` is **orthogonal** to FL aggregation algorithms, improving their performance in heterogeneous scenarios (Cifar100 with $\alpha = 0.5$ and Femnist with $\alpha = 0.01$). This shows that `FedGWC` and clustering are beneficial in this scenarios, also with personalization methods `pFedMe` (T Dinh et al., 2020) and `Per-FedAvg` (Fallah et al., 2020)

| FL method | Cifar100 | | Femnist | |
|---|---|---|---|---|
|  | No Clusters | `FedGWC` | No Clusters | `FedGWC` |
| FedAvg | $41.6 \pm 1.3$ | $\mathbf{53.4} \pm 0.4$ | $76.0 \pm 0.1$ | $\mathbf{76.1} \pm 0.1$ |
| FedAvgM | $41.5 \pm 0.5$ | $\mathbf{50.5} \pm 0.3$ | $83.3 \pm 0.3$ | $\mathbf{83.3} \pm 0.4$ |
| FedProx | $41.8 \pm 1.0$ | $\mathbf{49.1} \pm 1.0$ | $75.9 \pm 0.2$ | $\mathbf{76.3} \pm 0.2$ |
| pFedME | $93.4 \pm 0.1$ | $\mathbf{93.5} \pm 0.1$ | $63.6 \pm 0.3$ | $\mathbf{63.9} \pm 0.2$ |
| per-FedAvg | $89.0 \pm 0.1$ | $\mathbf{93.5} \pm 0.1$ | $71.2 \pm 0.2$ | $\mathbf{72.0} \pm 0.2$ |

As Figure 7 illustrates, `FedGWC` partitions the Cifar100 dataset into clients based on class distributions. Each cluster's distribution is distinct and non-overlapping, demonstrating the algorithm's efficacy in partitioning data with varying degrees of heterogeneity. In Figure 8, we report the domain detection on Cifar100, where 40 clients have clean images, 30 have noisy images, and 30 have blurred images. Table 4 shows that `FedGWC` performs a good clustering, effectively separating the different domains.

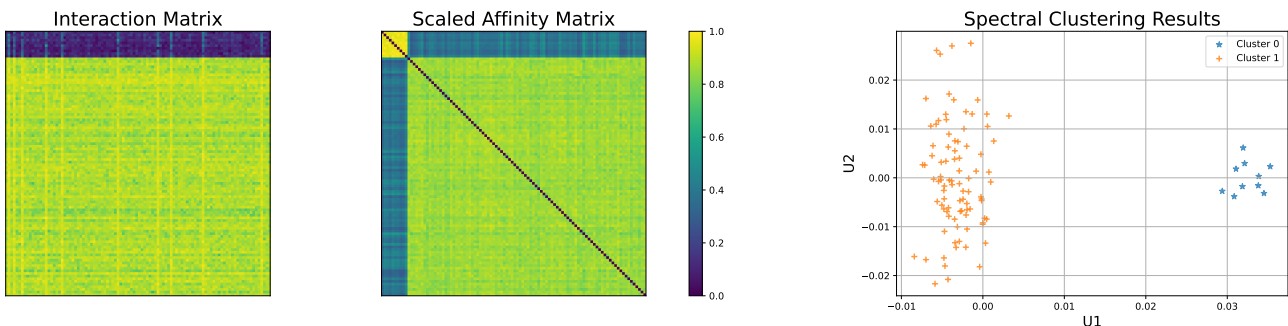

*Figure 5.* Homogeneous (Cifar10 $\alpha = 100$) vs heterogeneous clustering (Cifar10 $\alpha = 0.05$). The interaction matrix at convergence and the corresponding scaled affinity matrix are on the left. The scatter plot in the 2D plane with spectral embedding is on the right. It is possible to see that the algorithm perfectly separates homogeneous clients (orange) from heterogeneous clients (black)

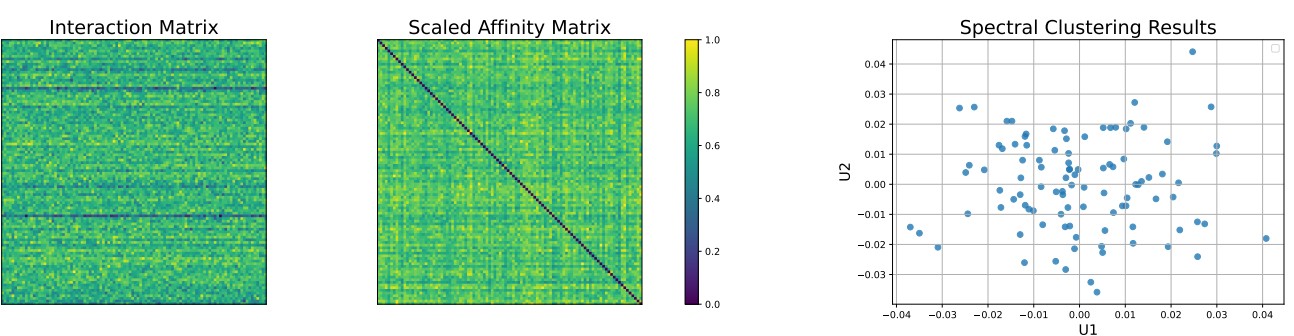

*Figure 6.* Homogeneous case (Cifar10 $\alpha = 100$). The interaction matrix at convergence and the corresponding scaled affinity matrix are on the left. The scatter plot in the 2D plane with spectral embedding is on the right. In the homogeneous case where no clustering is needed, *FedGW* does not split the clients.

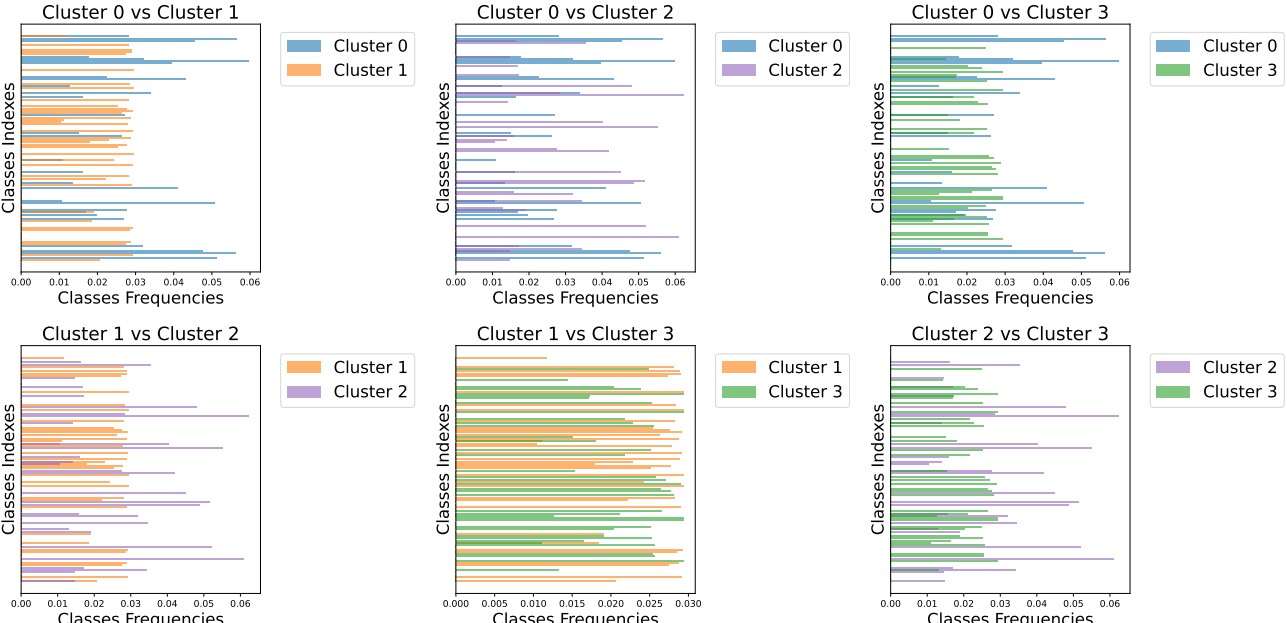

*Figure 7.* Class distributions among distinct clusters as detected by `FedGWC` on Cifar100. Specifically, we examine the class distributions for each pair of clusters, demonstrating that (1) the clusters were identified by grouping differing levels of heterogeneity and (2) there is, in most cases, an absence of overlapping classes.

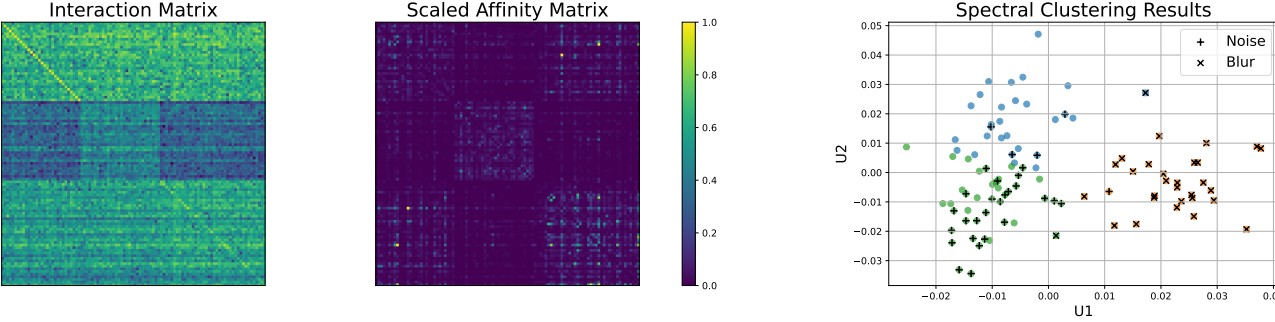

*Figure 8.* FedGWC in the presence of domain imbalance. Three domains on Cifar100: clean clients (unlabeled), noisy clients (+), and blurred clients (x). *Left*: is the interaction matrix $P$ at convergence from which it is possible to see client relations. *Center*: The affinity matrix $W$ computed with respect to the UPVs extracted from $P$, and on which FedGW_Clustering is performed. We can see that FedGWC clusters the clients according to the domain, as proved by results in Table 4.

