# OpenReview forum: "Interaction-Aware Gaussian Weighting for Clustered Federated Learning"
_ICML.cc/2025/Conference — ICML 2025 poster_

### Official Review · Reviewer_vYq3 · 2025-03-13

**Overall Recommendation:** 3

**Summary:**

This paper proposes a novel federated learning (FL) method called FedGWC (Federated Gaussian Weighting Clustering), which aims to mitigate the challenges of data heterogeneity and class imbalance in FL by clustering clients based on their data distributions. This method allows for the creation of more homogeneous client clusters, leading to more personalized and robust federated models.

**Claims And Evidence:**

Yes.

**Essential References Not Discussed:**

N/A.

**Experimental Designs Or Analyses:**

Yes.

**Methods And Evaluation Criteria:**

Yes.

**Other Comments Or Suggestions:**

N/A.

**Other Strengths And Weaknesses:**

Strengths:
1. The Gaussian reward mechanism provides a statistical method to determine the similarity between clients based on their empirical loss.
2. Comprehensive theoretical foundation and convergence guarantees.

Weakness:
1. Although the appendix includes experiments on additional datasets, the performance of baseline methods on these datasets is not provided.
2. The empirical loss may fail to fully capture the subtle differences in data distributions across different client datasets.
3. The authors claim that all clustering computations, including those based on interaction matrices and Gaussian weighting, are performed exclusively on the server. However, a detailed complexity analysis is needed for clarification.

**Questions For Authors:**

N/A.

**Relation To Broader Scientific Literature:**

N/A.

**Theoretical Claims:**

Yes.

---

### Official Review · Reviewer_3mtD · 2025-03-13

**Overall Recommendation:** 4

**Summary:**

This paper introduces FedGWC (Federated Gaussian Weighting Clustering), a clustered federated learning (CFL) framework designed to address data heterogeneity and class imbalance. The key idea behind FedGWC is to group clients into homogeneous clusters based on their data distributions, enabling personalized model training within each cluster. Key contributions of this paper include:

- Gaussian Weighting Mechanism: Clients are clustered by analyzing their empirical loss landscapes. A reward system quantifies alignment between client data distributions and cluster averages. Gaussian weights are introduced to keep track over time of these rewards, computed as a running average of the rewards.
- Interaction Matrix and Spectral Clustering: Pairwise client similarities (the Gaussian weights) are encoded in an interaction matrix, refined into an affinity matrix using RBF kernels. Spectral clustering partitions clients into groups, dynamically adjusting clusters based on convergence criteria of the interaction matrix.
- Wasserstein Adjusted Score: A novel metric, Kantorovich–Rubinstein metric, evaluates cluster cohesion under class imbalance, leveraging Wasserstein distance (with standard clustering quality metrics) to assess distributional alignment of ranked class frequencies.

The authors demonstrate that FedGWC outperforms existing CFL baselines (classical ones: IFCA, Sattler, and a recent one FedSem) and standard federated learning (FL) methods (FedAvg, FedProx) on benchmark datasets (Cifar100, Femnist) and large-scale real-world datasets (Google Landmarks, iNaturalist). Numerical results (Table 1 - 4) showed that FedGWC achieves higher accuracy and better clustering quality, except for the Femnist dataset, effectively handling domain shifts and class imbalance.

**Claims And Evidence:**

The majority of claims are supported by rigorous theoretical analyses and empirical (numerical experiments) evidence.

**Essential References Not Discussed:**

Essential references are well-discussed in the paper.

**Experimental Designs Or Analyses:**

The experimental design is largely sound for validating FedGWC’s core claims, with appropriate benchmarks and metrics. (See the last paragraph of the "Summary" section for details.) However, the authors did not provide a detailed analysis of why FedGWC performed far worse on the Femnist dataset, which is the simplest dataset, compared to the baselines.

**Methods And Evaluation Criteria:**

- Methods: Gaussian weighting (loss-based similarity), interaction matrix, and spectral clustering are well-suited for clustering clients in CFL.
- Evaluation: Standard FL benchmarks (Cifar100, Femnist) and large-scale datasets (Landmarks, iNaturalist) are appropriate for evaluating FedGWC's performance.
- Potential Limitations: the authors could consider more diverse datasets (e.g., NLP) to validate the generalizability of FedGWC.

**Other Comments Or Suggestions:**

- Why are the algorithms presented in the appendices rather than included in the main paper?
- Algorithm name capitalization in the References: e.g. Fldetector -> FLDetector. Use curly braces to enclose such terms in the bib file to preserve the capitalization.

**Other Strengths And Weaknesses:**

Strengths and weaknesses are discussed in previous sections.

**Questions For Authors:**

NA

**Relation To Broader Scientific Literature:**

The paper’s contributions advance clustered federated learning (FL) by addressing key limitations of prior work and successfully integrating insights from optimization, distribution alignment, etc.

P.S. I don't quite understand the exact meaning of "Broader Scientific Literature". I assume it refers to the broader context of the paper's contributions (mainly compared to existing methods in literature) in the whole field of federated learning.

**Theoretical Claims:**

Theorem 5.1 and Theorem 5.2 on the convergence of Gaussian weights are checked (proofs in Appendix A).

---

### Official Review · Reviewer_s2xj · 2025-03-17

**Overall Recommendation:** 2

**Summary:**

This paper focus on the clustered FL method to mitigate the non-iid problem in FL. FedGWC groups clients based on the data distribution. Gaussian reward mechanism is used to form homogeneous clusters. Comprehensive experiments demonstrate this method achieve better performance.

**Claims And Evidence:**

See weakness

**Essential References Not Discussed:**

The FL works referenced in related work are relatively outdated and new FL works should be added[R1-R4].

References:

[R1] Fan, Ziqing, et al. "Locally Estimated Global Perturbations are Better than Local Perturbations for Federated Sharpness-aware Minimization." *Forty-first International Conference on Machine Learning*.

[R2] Yang, Zhiqin, et al. "Fedfed: Feature distillation against data heterogeneity in federated learning." *Advances in Neural Information Processing Systems* 36 (2023): 60397-60428.

[R3] Lee, Taehwan, and Sung Whan Yoon. "Rethinking the flat minima searching in federated learning." *Forty-first International Conference on Machine Learning*. 2024.

[R4] Shi, Yujun, et al. "Understanding and mitigating dimensional collapse in federated learning." *IEEE Transactions on Pattern Analysis and Machine Intelligence* 46.5 (2023): 2936-2949.

**Experimental Designs Or Analyses:**

See weakness

**Methods And Evaluation Criteria:**

See weakness

**Other Comments Or Suggestions:**

See above

**Other Strengths And Weaknesses:**

- Concerns about the efficiency of this algorithm arise from the large number of training rounds required for it to converge.
- Why the author do not inquiry the data distribution from clients directly, rather infer by analyzing empirical loss function. Dose this induce privacy concern， if we can get the distribution of local data directly, what is the meaning of Gaussian Weighting Mechanism?
- Whether the transmission loss is an individual loss for each sample or a loss for each client？
- Is that the same for $\theta_{(1)}, \cdot,\theta_{(n_{cl})} $, for different groups.
-  It is unclear about how  to get $m^{t,s}$ in Equation (1).
- Why the Dirichlet parameter and client number is different for CIFAR-100 and Femnist, it is better to align the FL setting, it should be isolated from dataset selection. To be convincing is not an FL setting picked specifically for each dataset.

**Questions For Authors:**

No

**Relation To Broader Scientific Literature:**

Not relevant

**Theoretical Claims:**

See weakness

---

### Official Review · Reviewer_GoFZ · 2025-03-22

**Overall Recommendation:** 3

**Summary:**

This paper proposes FedGWC, a new clustered FL algorithm to tacke data heterogeneity and class imbalance among clients. FedGWC clusters clients based on their empirical losses, using a Gaussian reward mechanism. They also propose a new clustering metric, Wasserstein Adjusted Score, to evaluate cluster cohesion. The proposed algorithm is tested on benchmark datasets with standard partitions.

**Claims And Evidence:**

Yes. The algorithm is tested on benchmarking datasets.

**Essential References Not Discussed:**

No.

**Experimental Designs Or Analyses:**

Yes. I believe the experiment part is very solid. For datasets, the proposed algorithm uses both artificial-partitioned dataset and real federated datasets. The partition is also not designed for clustered FL. This is very different from many previous clustered FL papers demonstrating the genereralization of this algorithm. The author also compared the algorithm to important clustered FL baselines.

The soundness of the experiment can be further improved, if the author can compare to personalized FL baselines that are not restricted to clustered FL.

**Methods And Evaluation Criteria:**

Generally yes. It is intuitive to use empirical loss as signal for clustering. And the contruction of gamma in Subsection 4.1 is convincing.

However, I am a little bit confused about equation (3) in Subsection 4.2. In FL, typicall we want to choose $P_t$ large enough such that the aggregation is not significantly influenced by individual client's update.  I believe each $w_k^t$ measures the similarity from client $j$ to global, and is almost not relevant to client $j$ when the number of selected clients is large. it would be great if the author can show some evidence showing that the P matrix can capture client distribution similarity.

**Other Comments Or Suggestions:**

[1] is also a recent work of clustered FL considering both data quantity imbalance and non-IIDness. I suggest the author to discuss the difference between the proposed work and [1].

[1] Optimizing the Collaboration Structure in Cross-Silo Federated Learning. ICML 2023

**Other Strengths And Weaknesses:**

N/A

**Questions For Authors:**

N/A

**Relation To Broader Scientific Literature:**

No comments.

**Theoretical Claims:**

I briefly look through the statements and don't see any significant issues since they seem standard. However, I did not check the correctness of the proof.

---

### Decision · Program_Chairs · 2025-05-01

**Decision:**

Accept (poster)

**Comment:**

The paper presents an interesting approach. Even though there was no rebuttal, I feel the results are sufficiently interesting to be accepted.